# Anterior-posterior gradient of plasticity in primate prefrontal cortex

Mitchell R. Riley[1,3], Xue-Lian Qi[1], Xin Zhou[2] & Christos Constantinidis[1]

The functional organization of the primate prefrontal cortex has been a matter of debate with some models speculating dorso-ventral and rostro-caudal specialization while others suggesting that information is represented dynamically by virtue of plasticity across the entire prefrontal cortex. To address functional properties and capacity for plasticity, we recorded from different prefrontal sub-regions and analyzed changes in responses following training in a spatial working memory task. This training induces more pronounced changes in anterior prefrontal regions, including increased firing rate during the delay period, selectivity, reliability, information for stimuli, representation of whether a test stimulus matched the remembered cue or not, and variability and correlation between neurons. Similar results are obtained for discrete subdivisions or when treating position along the anterior-posterior axis as a continuous variable. Our results reveal that anterior aspects of the lateral prefrontal cortex of non-human primates possess greater plasticity based on task demands.

[1] Department of Neurobiology & Anatomy, Wake Forest School of Medicine, Medical Center Blvd, Winston-Salem, NC 27157, USA. [2] Department of Computer Science, Stanford University, Stanford 94305 CA, USA. [3] Present address: Department of Psychology, Vanderbilt University, Nashville, TN 37240, USA. Correspondence and requests for materials should be addressed to C.C. (email: cconstan@wakehealth.edu)

The lateral prefrontal cortex (PFC) plays a central role in higher cognitive functions including working memory, planning, and executive control[1]. Single neurons exhibit neural correlates of cognitive functions, typically representing multiple types of information and dynamically adjusting activity depending on the context of the current task[2–5]. Such neural correlates are presumed to be the effect of plasticity, emerging as a result of learning to perform a task and engaging in it, when circumstances demand[6]. Indeed, prefrontal neurons have been shown to be supremely plastic during learning of associations, categories, or rules[7–9].

Whether this plasticity varies across different prefrontal subdivisions has not been known. Evidence of learning-induced changes in task-related neuronal firing in primate PFC is scarce[10] and it is largely unknown if learning affects neuronal firing differentially across distinct areas of the prefrontal cortex. This question has broader implications for the organization of the prefrontal cortex, which has been a matter of debate, as some studies have documented clear functional specialization between areas[11,12] and others reported no sign of differentiation[21,22], possibly due to different training regimes. Anatomical evidence suggests a relative segregation of anatomical inputs from the posterior parietal into the dorsolateral prefrontal cortex and from the inferior temporal into the ventrolateral prefrontal cortex[13–15]. An anterior-posterior (AP) specialization has also been suggested based on anatomical and imaging studies, with more abstract operations localized anteriorly in the prefrontal surface[16–20]. Lesion studies support dissociable effects of dorsal and ventral, as well as posterior and anterior subdivisions, which agree with this specialization of function across prefrontal subdivisions[21,22]. Some neurophysiological evidence of dorsal–ventral and AP specialization has come from single-neuron recordings in animals naïve to training in cognitive tasks, which documented differences in selectivity for spatial and non-spatial information, and on progressive changes in response latency and receptive field size across the AP axis of the prefrontal cortex[23,24]. Yet, this result too appears at odds with the lack of specialization observed in other studies in trained monkeys.

We thus sought to investigate if all prefrontal areas are equally plastic as the result of training and if prefrontal cortex subdivisions maintain specialization after training. To this end, we recorded neurons from the same prefrontal areas before and after training in a spatial working memory task and determined changes in stimulus selectivity and task information. Our results show that training in the task induces changes differentially across prefrontal subdivisions, which in some respects abolishes functional differences between areas, though other functional characteristics persist, and the capacity for plasticity itself differentiates anterior and posterior areas.

## Results

**Study design.** Neurophysiological recordings were collected from a total of 3908 neurons in six monkeys, before training, and in four of the same animals after training in a working memory task (Supplementary Table 1 and Supplementary Figure. 1). We subdivided the lateral PFC into regions (Fig. 1a) and analyzed neurophysiological recordings obtained from five of these (posterior-dorsal region, $n = 182$ neurons pre-training/211 neurons post-training; mid-dorsal, $n = 756/690$; anterior-dorsal, $n = 359/222$; posterior-ventral, $n = 633/612$; anterior-ventral, $n = 122/121$). Some variations of the basic task were used in some sessions (variable duration of the delay period; sessions in which the sample stimulus was always match; sessions that displayed different shaped stimuli), resulting in slight variations of neurons available for some analyses (see Supplementary Methods). We

additionally projected the location of each recording onto the AP axis of the prefrontal cortex and used that as a continuous variable for our analysis (a value of 0 represents the genu of the arcuate; 1 the frontal pole). During experiments, we sampled neurons in an unbiased fashion by recording from all neurons encountered. Recordings were obtained primarily from the supragranular layers (see Supplementary Methods). The same stimuli (white squares appearing at 9 possible locations arranged on a $3 \times 3$ grid of 10° spacing between adjacent stimuli) were used in recordings prior to training in a working memory task, presented passively while the monkeys maintained fixation, and after training in the task (Fig. 1b, c).

**Neuronal activation.** Neural activity generally increased in the PFC following training (Fig. 2). This change was not uniform across task epochs and subdivisions. Average firing rate increased the most during the delay period, when the monkey was required to actively maintain information in memory; we note however that some neurons with persistent activity during the delay period were present even prior to training[25]. Firing rate also increased more in the anterior-dorsal PFC compared to the posterior-dorsal and mid-dorsal regions (Figs. 2, 3). A 2-way ANOVA on firing rates during the delay period, with dorsal PFC subdivisions and pre/post-training stage as factors revealed a significant main effect of training ($F_{1,2293} = 10.0$, $p = 0.0015$), as well as a significant interaction ($F_{2,2293} = 5.62$, $p = 0.0037$), confirming the differential effects of training across regions. The firing rate in the anterior-dorsal region was significantly higher than either the posterior or mid-dorsal regions after training (1-way ANOVA, $F_{2,1086} = 5.40$, $p = 0.005$; and Tukey post-hoc test, $p < 0.05$). The increase in anterior-dorsal activity in the delay period was due to two factors: first, a larger percentage of neurons was activated (10% before, 24% after training), i.e., exhibited significantly increased firing rate in the delay period over the fixation period (evaluated with a paired $t$-test, at the 0.05 significance level); and secondly, a higher firing rate was present among activated neurons (Fig. 3a, b). We rely on firing rate during the delay period for most subsequent analyses, since this was the interval where the greater training effects were observed. However we note that the training affected different areas in a systematic pattern in the baseline and cue periods as well (e.g. Supplementary Figure 2).

To ensure that changes in firing rate after training were not due to differences in experimental procedures, we used identical experimental and analytical methods in the two recording stages (see Methods). We additionally examined firing rates in subsets of neurons with excellent unit isolation, identified based on Signal-to-Noise Ratio > 5. This subset of neurons ($n = 2461$) exhibited the same differences in firing rates, including a greater increase in baseline activity in anterior and ventral areas after training (Supplementary Figure 3). Different prefrontal regions were also sampled in an interleaved fashion rather than sequentially. Therefore, the ages of monkeys were balanced across recordings in each area (Supplementary Table 2). The mean age of the subject in the pre-training stage was 6.7, 6.8, and 6.8 years, for the posterior-, mid-, and anterior-dorsal recordings, and 6.7 and 6.5 years for the posterior- and anterior-ventral recordings, respectively. Prefrontal responses are stable in monkeys after they reach this age range[26]. The mean age after training was 8.3, 8.5, and 8.2 years for recordings in the dorsal areas, and 8.6 and 8.9, for the ventral areas, respectively. For monkeys in which recordings were obtained both before and after training in the same region, a mean of 1.5, 1.7, and 1.6 years had elapsed, for the three dorsal regions.

Treating the recording location along the AP axis as a continuous variable confirmed increased firing rate changes in

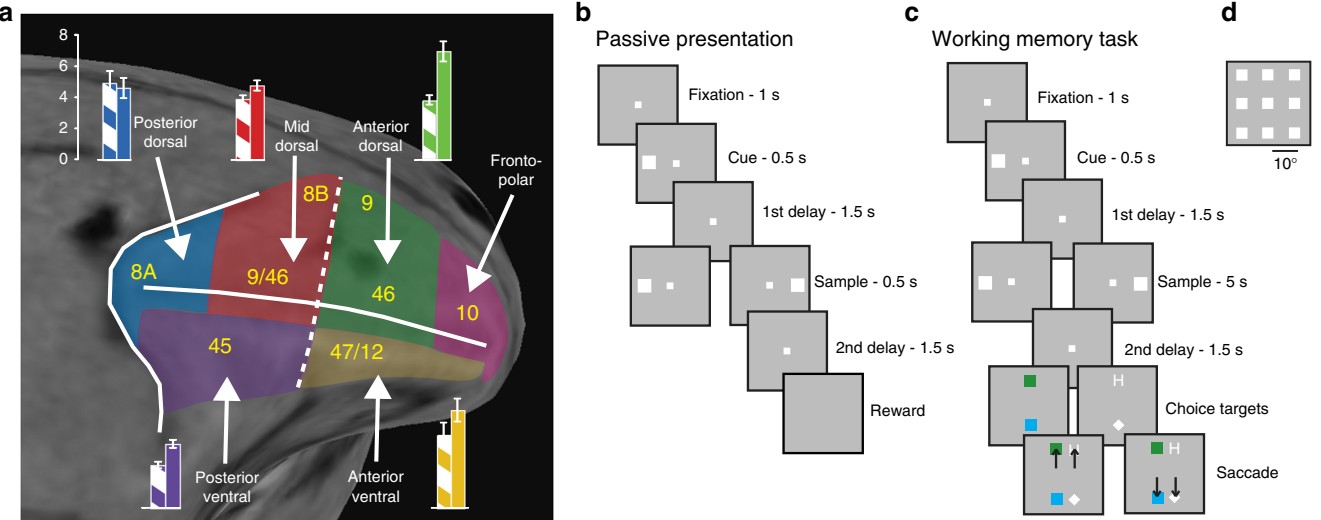

**Fig. 1** Diagram of PFC regions and Behavioral Paradigm. **a** The monkey prefrontal cortex was divided into regions as indicated. Insets represent average delay period firing rate (and sem) for each area, before (hatched bars) and after training (solid bars). **b** Sequence of events during the passive viewing paradigm. Monkeys were required to fixate a point and a stimulus was shown. After a delay period, a second stimulus was shown either matching the first in location or appearing at a different location. Another delay period followed, after which the monkeys were rewarded for maintaining fixation throughout the entire trial. **c** The spatial match/nonmatch working memory task. An additional display with two targets was displayed and the monkey now had to saccade to the green square/H if the two stimuli matched or the blue square/diamond if the two stimuli did not match. **d** Possible locations of the cue and sample stimuli

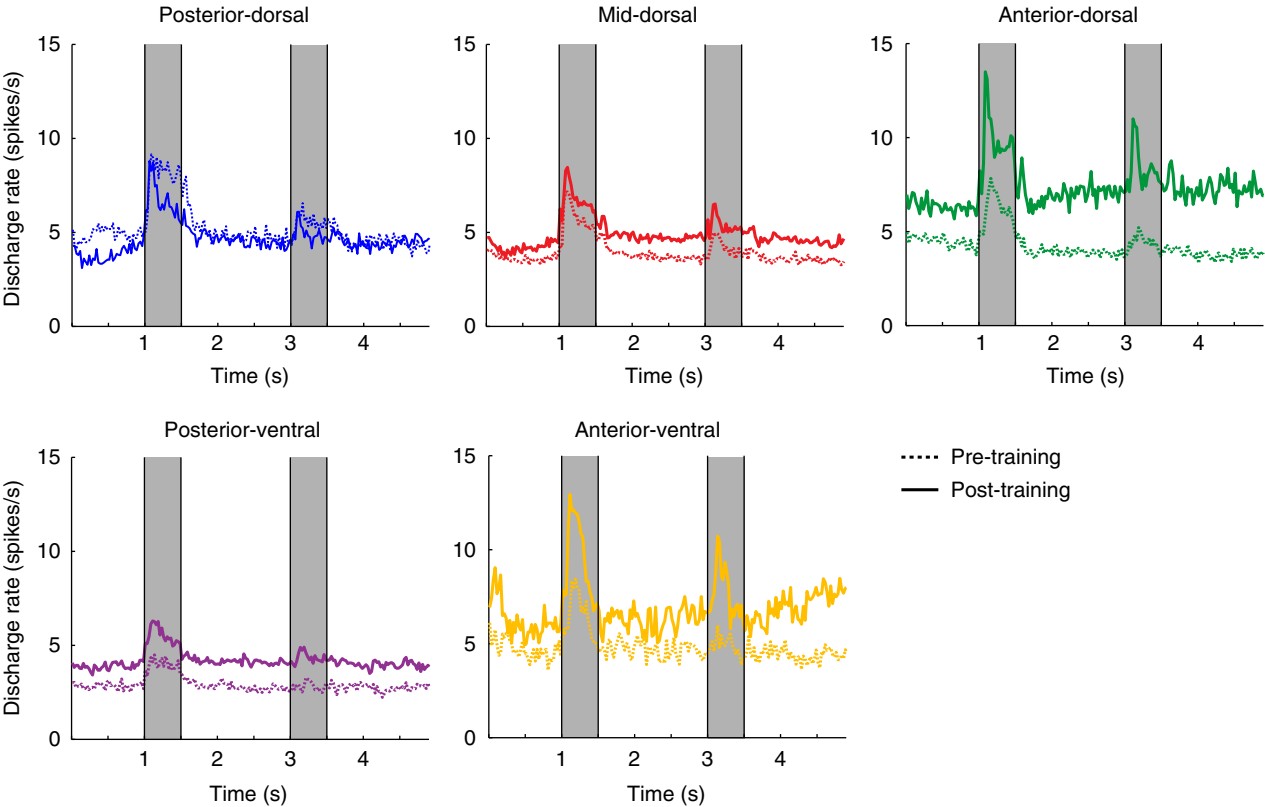

**Fig. 2** Firing rate changes after training. Peri-stimulus Time Histogram depicting mean firing rate during the trial, based on each neuron's best cue location, before and after training. Time 0 represents the onset of the fixation point. The cue stimulus is presented between 1.0–1.5 s (gray bar). Data are shown separately for each prefrontal region (Posterior-dorsal: Pre-training/post-training $n = 175/204$ neurons; Mid-dorsal $n = 668/659$; Anterior-dorsal $n = 298/212$; Posterior-ventral $n = 502/581$; Anterior-ventral $n = 92/117$)

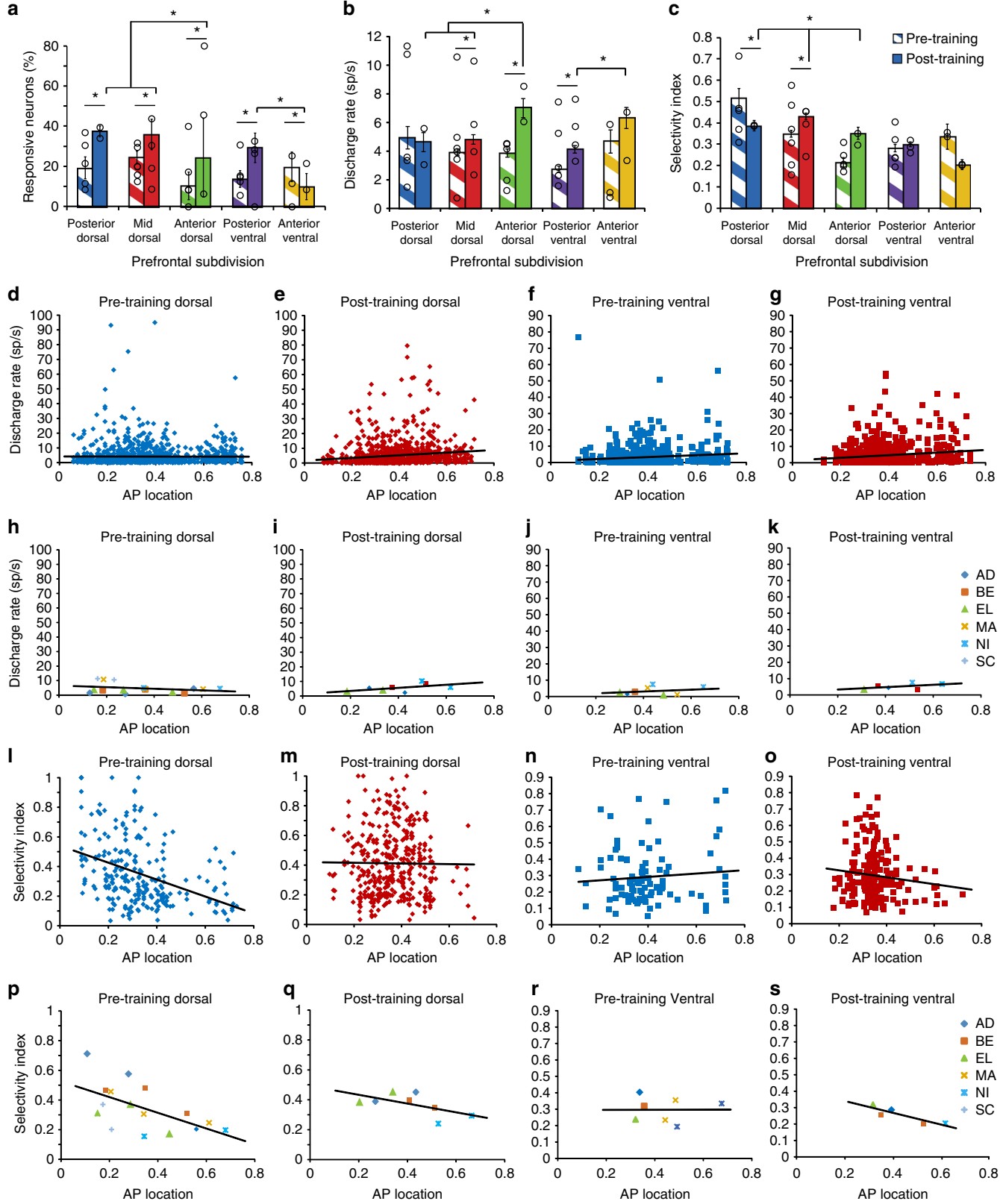

more anterior parts of the prefrontal cortex (Fig. 3d, e). Prior to training, firing rate did not vary appreciably as a function of dorsal AP location (Fig. 3d), as evidenced by a slope of a regression line that was not significantly higher than zero ($F_{1,1206} = 0.03$, $r = 0.002$, $p = 0.872$). On the contrary, after training (Fig. 3e) the slope of the regression line was highly

significant ($F_{1,1082} = 17.83$, $r = 0.179$, $p = 2.6 \times 10^{-5}$), as was the increase of the post-training slope relative to the pre-training one (F-test, $F_{1,2288} = 15.31$, $p = 0.0001$). Among ventral regions (Fig. 3b, f, g), increases in firing rate were also greater in the anterior than the posterior division after training. Increases in activity were not exclusive to the delay period; cue responses and

**Fig. 3** Responsiveness changes after training. **a** Bar graphs represent the percentage of neurons that exhibited significant responses in the delay period over the baseline period in each prefrontal region ($n = 3908$ total neurons). Superimposed circles represent percentages calculated separately for each individual monkey. Unequal sample sizes were obtained from each monkey therefore bars do not correspond to the mean of the individual monkey values. **b** Mean firing rate (and SEM) during the delay period of the task. The best stimulus for each neuron was used (Posterior-dorsal: Pre-training/post-training $n = 178/215$ neurons; Mid-dorsal $n = 719/662$; Anterior-dorsal $n = 313/212$; Posterior-ventral $n = 610/586$; Anterior-ventral $n = 92/117$). Statistical comparisons between groups are based on 1-way ANOVA between areas, before and after training, and 2-way ANOVA for all areas, combined; asterisk indicates results of Tukey post-hoc test ($p < 0.05$). Superimposed circles represent means calculated separately for each individual monkey. **c** Average SI values of delay period activity, before and after training. For this analysis, we only used neurons that responded significantly during the delay period (Posterior-dorsal: Pre-training/post-training $n = 34/77$ neurons; Mid-dorsal $n = 174/253$; Anterior-dorsal $n = 33/52$; Posterior-ventral $n = 84/173$; Anterior-ventral $n = 16/11$). Superimposed circles represent means calculated separately for each individual monkey. **d–g** Scatterplots show mean delay period firing rate for all neurons arranged by their relative position along the AP axis, for pre-training dorsal (**d**), post-training dorsal (**e**), pre-training ventral (**f**), and post-training ventral (**g**) groups. Line represents linear regression. **h–k** Each point represents the mean value of delay period discharge rate calculated in a region, from one monkey. The point is plotted in the AP location corresponding to the mean value of the position of neurons used in the mean. Line represents linear regression through these monkey-derived means. **l–o** Scatterplots show selectivity index values during the delay period for significantly responding neurons organized by their AP position. Neurons are shown for pre-training dorsal (**h**), post-training dorsal (**i**), pre-training ventral (**j**), and post-training ventral (**k**) groups. **p–s** Each point represents the mean value of selectivity index calculated in a region, from one monkey

fixation activity was also generally elevated after training, throughout the PFC (Fig. 3 and Supplementary Figure 2). An important issue in these comparisons was whether changes were consistent across individual subjects. To address this question, we computed the mean firing rate separately for each monkey and area. We then plotted this rate on a position that corresponded to the mean AP location of neurons sampled (Fig. 3h–k). The regression lines obtained in this fashion were virtually identical to those determined based on all neurons and confirmed an increase in the slope of the regression line after training (see also Supplementary Methods).

**Stimulus responsiveness, discriminability, and information**. Increased activation of a region does not necessarily imply increased responsiveness to stimuli. We therefore quantified the range of firing rates exhibited for stimuli appearing at different spatial locations, in neurons that were responsive during at least one epoch of the trial. We defined a selectivity index (SI) as (Max − Min)/(Max + Min) where Max and Min represent the firing elicited after appearance of the best and worst stimulus, calculated by averaging firing rate during the delay period of the task. The SI value essentially normalizes firing rate and weighs low firing neurons no less than high firing ones. This analysis revealed differential effects of training on areas across the AP axis of the dorsal PFC (Fig. 3c). Prior to training, delay-period selectivity index values differed significantly between the dorsal regions (ANOVA, $F_{2,238} = 17.6$, $p = 7.4 \times 10^{-8}$), with the posterior-dorsal region being most selective. Training induced the greatest increase in selectivity in the anterior-dorsal region, amounting to a mean SI value increase of 0.14 for the anterior region, 63% relative to its pre-training value (Fig. 3). In contrast, the post-training selectivity of the posterior-dorsal region decreased relative to its pre-training value. The mid-dorsal region exhibited a more modest increase, amounting to an SI value increase of 0.08, 38% over its pre-training value. A 2-way ANOVA on delay-period SI values, with dorsal PFC subdivisions and pre/post-training stage as factors, revealed a significant interaction ($F_{2,599} = 10.98$, $p = 2.07 \times 10^{-5}$), confirming the differential effects of training across regions. Comparing specifically the mid-dorsal and anterior-regions, confirmed a significantly higher increase in SI values for the anterior-dorsal region (permutation test, evaluated at a = 0.05).

Treating location along the AP axis as a continuous variable (Fig. 3h, i) revealed a significantly decreasing slope prior to training (linear regression, $F_{1,239} = 41.41$, $r = 0.384$, $p = 6.7 \times 10^{-10}$) but a non-significantly decreasing one after training ($F_{1,362}$

= 0.05, $r = 0.011$, $p = 0.830$). The slopes corresponding to the two stages were significantly different from each other (F-test, $F_{1,601} = 14.59$, $p = 0.0001$). A negative slope was also evident for the dorsal prefrontal cortex based on individual monkeys (Fig. 3p); this slope became less negative after training (Fig. 3q). Similar results were obtained during the cue presentation period, for responsive neurons during that task epoch (Supplementary Figure 4). Among the ventral areas, we observed mixed changes in selectivity; no significant difference in the posterior-ventral area but a decrease in the anterior-ventral, both in the delay (Fig. 3c) and in the cue period (Supplementary Fig. 4). The anterior-ventral PFC exhibited the greatest change after training, in this fashion (Fig. 3c, n, o, r, s).

The ability to discriminate between stimuli does not only depend on mean firing rates, but also on their reliability[27]. To quantify the reliability of stimulus discrimination, we performed a Receiver-Operating-Characteristic (ROC) analysis. We determined each neuron's best and worst stimulus location based on the stimulus presentation period and quantified the ability to discriminate between the firing rate distributions obtained at each time point (Fig. 4). The analysis largely confirmed the findings of the firing rate comparisons. Prior to training, a gradient of decreasing selectivity in the delay period was evident from the posterior to anterior dorsal areas (Fig. 4a). A regression analysis of individual neuron ROC values on AP position (Fig. 4d) indicated a significantly negative slope ($F_{1,1140} = 18.4$, $r = 0.126$, $p = 1.97 \times 10^{-5}$). This was abolished after training (Fig. 4b, c), so that a regression analysis revealed no significant effect after training ($F_{1,759} = 0.54$, $r = 0.023$, $p = 0.46$, Fig. 4e). In the ventral prefrontal cortex, slightly positive slopes were present both before ($F_{1,593} = 5.9$, $r = 0.099$, $p = 0.015$) and after training ($F_{1,544} = 4.44$, $r = 0.09$, $p = 0.035$, Supplementary Fig. 5).

Information that can be represented in neuronal firing rates about the stimulus location is only partially captured by ROC analysis, which compares two stimulus conditions. We therefore relied on Mutual Information, which determines how well stimulus location can be separated from other locations based on firing rate of all locations[27,28]. Prior to training, a regression of Mutual Information values on AP position (Fig. 5a) indicated no significant slope ($F_{1,1140} = 0.011$, $r = 0.003$, $p = 0.916$). In contrast, this slope was significantly positive after training, suggesting that more anterior neurons represented higher levels of information, increasing more after training ($F_{1,1074} = 44.48$, $r = 0.2$, $p = 3.99 \times 10^{-11}$, Fig. 5b). In the ventral PFC, positive slopes were present both before ($F_{1,593} = 12.9$, $r = 0.146$, $p = 0.0004$) and after training ($F_{1,697} = 11.41$, $r = 0.127$, $p = 0.0008$), with overall levels of information increasing after training (Fig. 5c, d). A

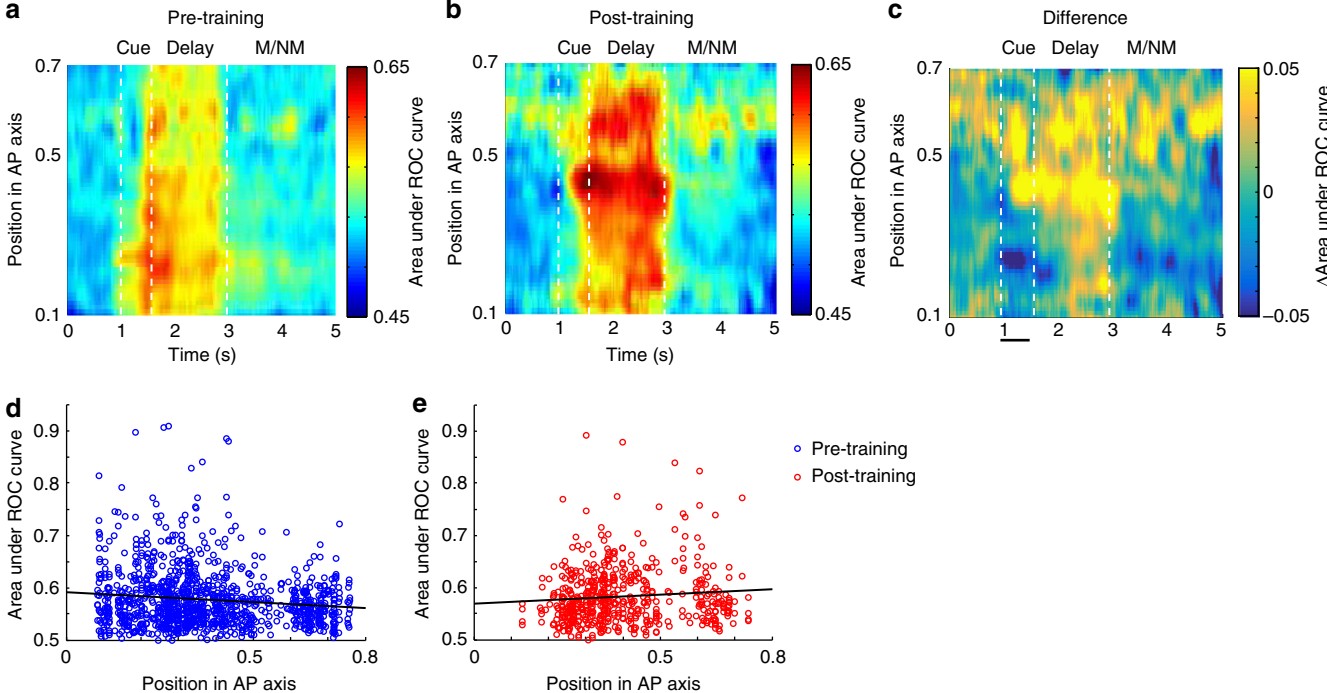

**Fig. 4** Receiver operating characteristic analysis. **a, b** ROC values are plotted along the Anterior-Posterior axis of the dorsal prefrontal cortex before and after training (n = 3040). Each ROC value was obtained in a 250 ms bin across time, and at across the AP axis, and then smoothed with a Gaussian. Cue presentation is indicated between horizontal lines spanning the time interval of 1.0–1.5 s in the plot. First delay period spans the interval of 1.5–3.0 s. This is followed by the sample stimulus (both match and nonmatch pooled together). **c** Difference of pre- and post-training stages. **d** Each point represents the mean ROC value of the 250 ms bins tiling the delay period of each individual neuron, prior to training. Line represents linear regression. **e** ROC values of each neuron, during the delay period, in recordings obtained after training. Total sample size, n = 3040 neurons

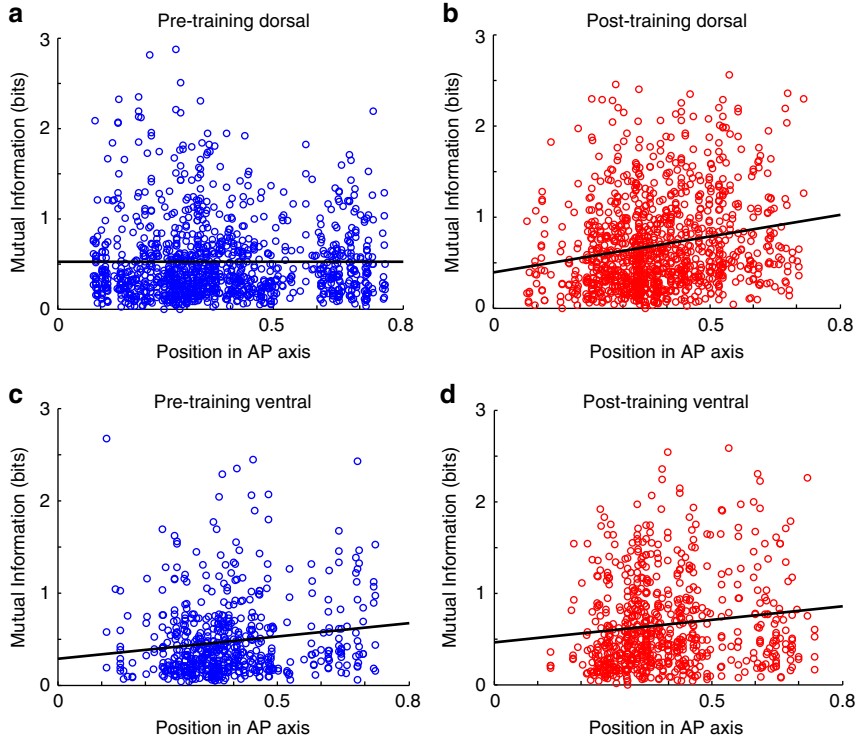

**Fig. 5** Mutual information analysis. **a** Mutual Information values based on spike counts in the 1.5 s of the delay period are plotted along the Anterior-Posterior axis of the dorsal prefrontal cortex before and after training. Each point represents the Mutual Information value of one neuron, prior to training. Line represents linear regression. **b** Mutual Information values in the ventral prefrontal cortex. **c–d** Mutual information values after training. Total sample size, n = 3508 neurons

similar pattern of Mutual Information changes was obtained in the cue period (Supplementary Figure 6).

**Variability and correlation**. In addition to modulating the mean firing rate of neuronal responses, it is increasingly recognized that cognitive factors also affect the trial-to-trial variability and correlation of neuronal discharges[29,30]. We wished therefore to determine how the variability of neuronal responses (quantified based on the Fano factor i.e., variance of spike counts divided by their mean), and the correlation of firing rates between simultaneously recorded neurons (quantified as the spike-count correlation) differed between areas and was affected by training and performance of a working memory task. Prior to training, Fano factors of spike counts during the delay period were generally higher for more anterior areas (Fig. 6a, b). A regression analysis of (the logarithm of) Fano factor on AP position revealed a significantly positive slope in both the dorsal ($F_{1,1139} = 8.72$, $r = 0.087$, $p = 0.003$) and ventral PFC ($F_{1,592} = 7.04$, $r = 0.108$, $p = 0.008$). This result is consistent with the findings of lower stimulus responsiveness and selectivity in anterior areas. When an identical stimulus is presented repeatedly, neurons more closely tied to stimulus representation, in posterior areas, would be expected to exhibit low variability. On the other hand, neurons influenced to a greater extent by cognitive factors such as attention and motivation would be expected to be more sensitive to (unobserved) variations of these internal variables from trial-to-trial, even though the stimulus conditions are exactly the same.

After training, this gradient of variability was exaggerated. The mean Fano factor slightly decreased in the posterior-dorsal region but increased in the anterior dorsal (Fig. 6c). As a result, the slope of the regression line was steeper ($F_{1,1073} = 46.16$, $r = 0.203$, $p = 1.8 \times 10^{-11}$). A similar increase in regression slope was also

observed (Fig. 6d) in the ventral PFC ($F_{1,696}$, $r = 0.233$, $p = 4.5 \times 10^{-10}$).

Spike-count correlation, which reflects variability shared between neurons recorded simultaneously, followed a similar pattern (Fig. 6e–g). This measure captures the correlation of firing rate deviations around the mean following an identical stimulus presentation across trials[29]. Theoretical and experimental findings suggest that such correlations are primarily the effects of cognitive factors (such as attention or motivation), which affect jointly the neurons under study, and may vary between trials, causing firing rate to shift in unison[31,32]. A data set of 3402 pairs of neurons recorded simultaneously before training and 1266 after training was available. A regression analysis of spike counts in the delay period on AP position revealed a significant positive slope in dorsal prefrontal cortex prior to training ($F_{1,2237} = 9.42$, $r = 0.064$, $p = 0.002$), suggesting a greater degree of co-variations of firing rate around the mean in anterior areas. The slope more than doubled after training (from 0.07 to 0.15), while being significantly different than zero ($F_{1,834} = 14.1$, $r = 0.129$, $p = 1.8 \times 10^{-4}$). For the ventral prefrontal cortex (Fig. 6g, h), the slope of the regression curve failed to reach significance prior to training ($F_{1,1816} = 3.2$, $r = 0.052$, $p = 0.07$) but was statistically significant after training ($F_{1,660} = 15.1$, $r = 0.184$, $p = 1.15 \times 10^{-4}$). The results of the Fano factor and spike-count correlation analysis support the idea that anterior areas are affected to a greater extent by cognitive factors, after training.

**Match/nonmatch responses**. Anterior prefrontal neurons were also more plastic in representing information about the second major component of performing the task, namely judging whether the second stimulus was matching the first or not. Prefrontal neurons are known to exhibit selectivity for the match or

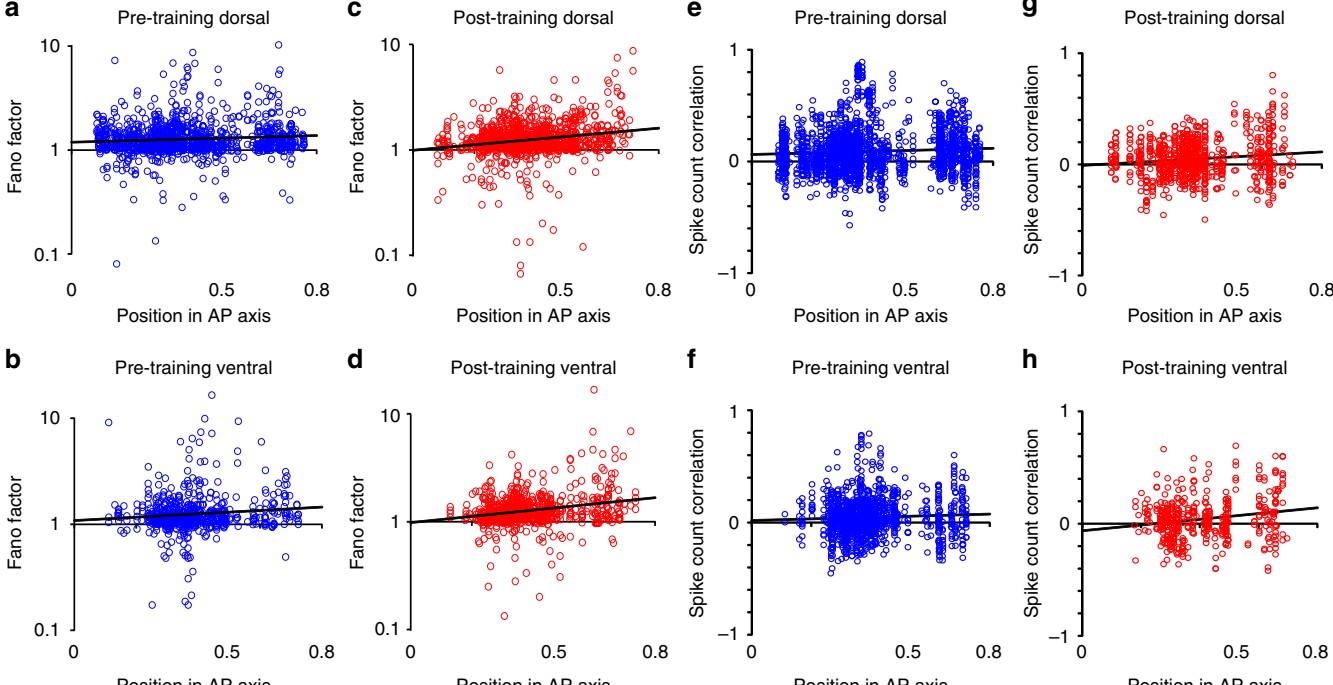

**Fig. 6** Variability and correlation. **a** Fano factor values based on spike counts in the delay period are plotted along the anterior-posterior axis of the dorsal prefrontal cortex before training. Line represents linear regression. **b** Fano factor values in the ventral prefrontal cortex. **c, d** Fano factor values after training. Total sample size in panels (**a–d**): n = 3508 neurons. **e** Spike-count correlation values computed during the delay period are plotted along the anterior-posterior axis of the dorsal prefrontal cortex before training. **f** Spike-count correlation in the ventral prefrontal cortex prior to training. **g, h** Spike-count correlation values after training. Total sample size in panels (**e–h**), n = 4668 pairs of simultaneously recorded neurons from separate electrodes, spaced 0.5–1 mm apart

nonmatch status of a stimulus, an effect predictive of behavioral performance[33,34]. We wished to examine how this task-driven selectivity was incorporated in different prefrontal subdivisions. We therefore plotted the responses to the best second stimulus when it was preceded by the same stimulus (and constituted a match), and when it was preceded by a stimulus at a different location (and constituted a nonmatch). For just over half of the neurons, responses were lower for a stimulus appearing as a nonmatch; for the remaining neurons the reverse was true. This difference in firing rate, which is a measure of match/nonmatch selectivity for each neuron, became greater after training, as expected, since monkeys were explicitly trained to make a match/nonmatch judgement (Supplementary Figure 7). Among dorsal subdivisions, this increase was proportionally greatest in the anterior-dorsal region. A 2-way ANOVA, with dorsal subdivisions and pre/post-training stage as factors revealed a significant main effect of stage for nonmatch-preferring neurons ($F_{1,986} = 19.72$, $p = 9.51 \times 10^{-4}$) as well as an interaction ($F_{2,986} = 7.01$, $p = 0.001$), indicating that training had different effects across areas. The effects of training were similarly greater in the anterior-ventral compared to the posterior-ventral region: main effect of stage ($F_{1,570} = 8.69$, $p = 0.0033$); interaction ($F_{1,570} = 10.25$, $p = 0.0014$). This was also true for match-preferring neurons in both dorsal (stage: $F_{1,944} = 6.66$, $p = 0.01$; interaction: $F_{2,944} = 3.38$, $p = 0.03$) and ventral regions (stage: $F_{1,592} = 13.87$, $p = 0.0002$; interaction: $F_{1,592} = 7.05$, $p = 0.0082$).

The absolute difference of match and nonmatch firing rate |M-NM| revealed most clearly the differences between areas (Fig. 7a). The highest firing rate representing the match or nonmatch status of a stimulus was represented in anterior areas after training, though little difference was present between areas, prior to training. Expressing the match–nonmatch difference as d′ produced a very similar pattern (Fig. 7b). Treating neuron location across the AP axis as a continuous variable also confirmed that greater increases in match-nonmatch responses were evident at more anterior locations (Fig. 7c, d). Prior to training, the absolute value in firing rate difference between match and nonmatch stimuli did not depend significantly on AP position for either dorsal (regression analysis, $F_{1,1182} = 0.004$, $p = 0.948$) or ventral prefrontal cortex ($F_{1,640} = 0.003$, $p = 0.956$). After training, a significantly positive slope was present in both dorsal ($F_{1,1943} = 25.32$, $p = 6.06 \times 10^{-7}$) and ventral PFC ($F_{1,543} = 20.46$, $p = 7.49 \times 10^{-6}$). Regression slopes obtained from all neurons (Fig. 7c, d, g, h) were very similar to those obtained when we relied on match rates computed separately in each monkey (Fig. 7e, f, I, j; see also Supplementary Methods).

**Relationship between activity and behavior.** Specialization of functional properties between different prefrontal subdivisions, taken to the extreme, may imply that execution of the working memory task after training may depend exclusively on the activity of only the anterior regions. To test this idea, we compared neural activity in correct and error trials in each region, following appearance of the cue at the best location of each neuron. The area under the ROC curve comparing the distribution of firing rates in correct and error trials involving the same stimulus and only differing in terms of the monkey's choice is referred to as Choice Probability[35]. ROC values in the delay period tended to be greater than chance (0.5) in all prefrontal subdivisions (Supplementary Fig. 8). This was the result of delay-period firing rates being greater in correct than error trials (Supplementary Figure 9). In other words, trials in which activity was lower in the delay period were more likely to result in errors, and this was true for all prefrontal subdivisions. A comparison of ROC values computed over the entire delay period revealed no significant

difference between dorsal subdivisions (1-way ANOVA, $F_{2,338} = 2.15$, $p > 0.1$) or ventral subdivisions (1-way ANOVA, $F_{1,221} = 2.25$, $p > 0.1$).

We similarly examined Choice Probability values, when the second stimulus appeared at the neuron's preferred location and the monkey determined whether it was a match or nonmatch. Again, we saw values that were greater than 0.5 in all prefrontal subdivisions (Supplementary Figure 7B-C and 8). For neurons with overall greater responses to nonmatch stimuli, that meant that responses to nonmatches were greater in correct than error trials. The reverse was true for match-preferring neurons. A comparison of ROC values computed over the second stimulus period again revealed no significant difference between dorsal subdivisions (1-way ANOVA, $F_{2,105} = 1.89$, $p > 0.1$ for nonmatch preferring; $F_{2,118} = 2.01$, $p > 0.1$ for match-preferring neurons). No significant difference was present for ventral subdivisions (1-way ANOVA, $F_{1,79} = 0.31$, $p > 0.5$ for nonmatch preferring; $F_{1,72} = 0.24$, $p > 0.6$ for match preferring). These results demonstrate that errors in the task were associated with changes in activity of all prefrontal subdivisions, during the period of maintenance of the first stimulus in memory, and during the judgement of whether the second stimulus constituted a match or not.

**Effects of training and of task execution.** In the second stage of recordings, the monkeys had both been trained to perform the working memory task and they executed the task. The differences we observed across prefrontal areas between stages could be due to greater, long-term changes that anterior areas underwent over the period of training, or due to flexible, dynamic changes that anterior areas were more amenable to when the monkeys executed the task, compared to viewing the same stimuli while passively fixating. To distinguish between these two possibilities, we collected an additional data set from two monkeys after training, while they viewed the stimuli passively using the exact same stimulus conditions as prior to training, and compared responses of the same neurons during passive presentation and during active performance of the spatial working memory task. In these experiments we focused primarily in more anterior and ventral areas, where the greatest pre- vs. post-training changes were observed. We performed this experiment in a subset of sessions after training. Since data from all sessions were not available, we primarily focused on neurons that exhibited significant responses for any of the stimuli and trial epochs and were tested with both passive presentation and during active task performance (because neurons not responding to any stimulus in our set were less likely to be informative about whether passive presentation or active task execution affects their responses). This yielded a sample of 187 neurons (mid-dorsal, $n = 94$ neurons; anterior-dorsal, $n = 24$; posterior-ventral, $n = 22$; anterior-ventral, $n = 47$). We also compared these neurons to a sample of neurons recorded prior to training that exhibited significant responses, identified in the same fashion (mid-dorsal, $n = 368$ neurons; anterior-dorsal, $n = 153$; posterior-ventral, $n = 178$; anterior-ventral, $n = 46$). Data from the entire available sample of 358 neurons recorded during passive presentation after training and compared with all 1652 neurons recorded in the same areas prior to training are shown in Supplementary Figure 10.

The effects of active task performance vs. passive presentation were generally minor compared to the pre- vs. post-training comparisons. We first considered the difference in match and nonmatch rates, comparing |M-NM| values obtained during the second stimulus presentation (Fig. 8a). Only the anterior-dorsal area exhibited a significant effect of task performance vs. passive presentation after training (1-tailed paired $t$-test, $t_{23} = 1.95$, $p = 0.032$). Differences of active vs. passive presentation represented a change of 15%,

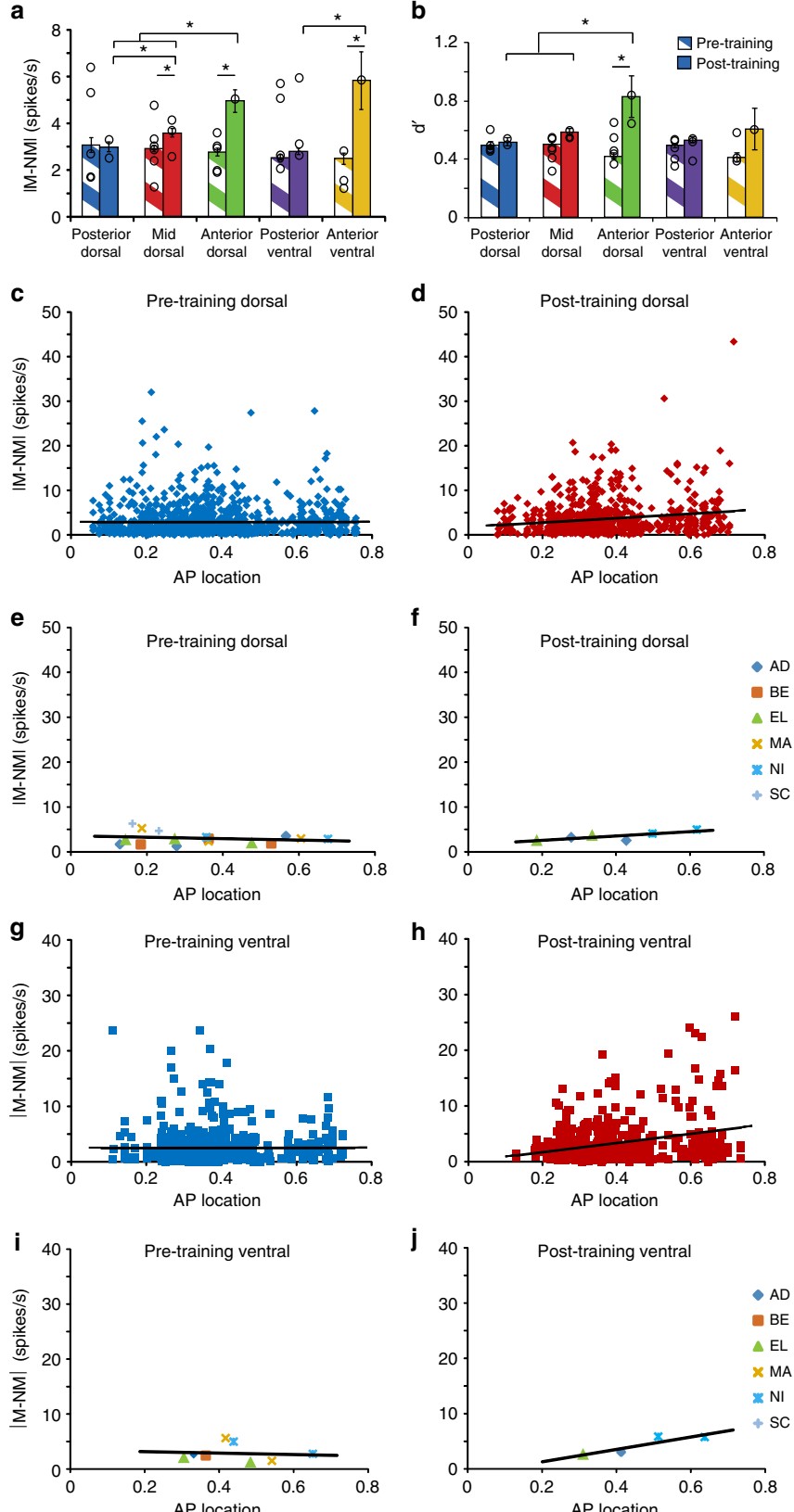

53%, −10 and −13% for the mid-dorsal, anterior-dorsal, posterior-ventral, and anterior-ventral areas, respectively. In contrast, when we compared passive presentation of stimuli recorded before training and after training, significant increases were present for the mid-dorsal, posterior-ventral, and anterior-ventral areas (1-tailed $t$-test,

mid-dorsal: $t_{460} = 5.01$, $p = 7.8 \times 10^{-7}$; anterior-dorsal: $t_{175} = 1.75$, $p = 0.08$; posterior-ventral: $t_{198} = 2.07$, $p = 0.039$; anterior-ventral: $t_{91} = 3.57$, $p = 5.81 \times 10^{-4}$). Differences of pre- vs. post-training values of passive sessions represented changes of 157%, 46%, 99% and 123% for the four areas, respectively.

**Fig. 7** Match–nonmatch difference. **a** Mean absolute firing rate difference during the sample (second stimulus) presentation period for a stimulus at the best location of each neuron, when it appears as a match relative to when it appears as a nonmatch. Error bars represent the standard error of the mean calculated across all neurons, before and after training. Only neurons that were tested with the Match-Nonmatch presentation were used in this analysis ($n = 3132$). Stars indicate significant differences ($p < 0.05$) in 1-way ANOVA comparing responses between areas prior to training; and 2-way ANOVA and post-hoc Tukey test comparing responses in the same area before and after training. Superimposed circles represent means calculated separately for each individual monkey. Unequal sample sizes were obtained from each monkey therefore bars do not correspond to the mean of the individual monkey values. **b** The same match/nonmatch difference for the same population of neurons is expressed as d'. **c** Match–nonmatch discharge rate difference plotted along the AP axis of the dorsal prefrontal cortex before training. Each point represents the value of one neuron, prior to training. Line represents linear regression. **d**. Match–nonmatch discharge rate difference in the dorsal prefrontal cortex, after training. **e**, **f** Each point represents the mean value of the |M–NM| quantity calculated in a region, from one monkey, for the dorsal prefrontal cortex. The point is plotted in the AP location corresponding to the mean value of the recordings. Line represents linear regression through these monkey-derived means. **g**, **h** Match–nonmatch discharge rate difference in the ventral prefrontal cortex, before and after training. **i**, **j** Data from individual monkeys plotted as in (**d**, **e**) for the ventral prefrontal cortex

Measures of stimulus information were also relatively stable in passive and active sessions recorded after training (Fig. 8b). Mutual Information values were significantly higher during the active presentation after training for the mid-dorsal (1-tailed paired $t$-test, $t_{93} = 1.69$, $p = 0.047$) and posterior-ventral regions ($t_{21} = 2.53$, $p = 0.010$). In contrast, all four areas exhibited significant increases in mutual information values during passive presentation post-training compared to pre-training (1-tailed $t$-test, mid-dorsal: $t_{460} = 12.0$, $p = 2.1 \times 10^{-29}$; anterior-dorsal: $t_{175} = 3.91$, $p = 6.4 \times 10^{-5}$; posterior-ventral: $t_{198} = 4.88$, $p = 1.08 \times 10^{-6}$; anterior-ventral: $t_{91} = 3.84$, $p = 1.13 \times 10^{-4}$). Again, Mutual Information changes were proportionally greater for the pre-training vs. post-training passive presentation comparison, than the post-training passive vs. active presentation comparison (Fig. 8b).

Neuronal firing variability was the only measure that exhibited comparable differences between passive and active presentation than it did between pre- and post-training. Mean values of Fano factor during active presentation were significantly higher than passive presentation after training for the anterior-dorsal area (1-tailed paired $t$-test, $t_{23} = 1.89$, $p = 0.035$). Fano factor values were also significantly higher for the post-training passive condition compared to the pre-training passive condition in the mid-dorsal area (1-tailed $t$-test, $t_{460} = 4.84$, $p = 8.7 \times 10^{-7}$).

## Discussion
An AP organization of the primate prefrontal cortex been suggested based on anatomical results revealing a hierarchical organization of prefrontal inputs and outputs[22,36,37]. Similarly, human imaging studies have placed representation of stimulus properties posteriorly, and more complex operations anteriorly[17,38,39]. However, previous neurophysiological studies have reported little difference in properties between areas, but rather robust representation of variables relevant for the task, regardless of which subdivision has been sampled[2–5]. Here, we document that AP areas differ not only in their patterns of responses to stimuli, but also to the degree of plasticity they exhibit after a spatial working memory task is learned. Firing rate during the delay period after the stimulus presentation, selectivity, reliability, information, and representation of the match or nonmatch status of a stimulus increased disproportionately in the anterior PFC after training. Trial-to-trial variability and spike-count correlation among neurons were also more pronounced in the anterior PFC. All monkeys were trained in a spatial working memory task and it is upon future studies to determine if similar effects would be observed for other cognitive tasks.

A previous neurophysiological study focusing exclusively on recordings prior to training, described an AP hierarchy in terms of response latency, receptive field size, and selectivity for stimuli[23]. We now report that many of the properties of prefrontal neurons were radically altered after training to perform a working memory task. In some instances, these changes erased the apparent specialization between areas, e.g., in terms of selectivity for the stimuli (Fig. 3), in line with previous studies that have failed to detect regional specialization in trained animals[40,41]. Nonetheless, some regional differences in functional properties persisted (e.g., in terms of responsiveness in the delay period activity, trial-to-trial variability, and spike-count correlation), and new differences emerged (e.g., in match/nonmatch representation). Prefrontal specialization was also starkly evident in that effects of training were localized disproportionately in the anterior areas.

Neuronal activity generally increased after training, particularly during the delay period. This result is in agreement with human imaging studies that have reported increases in activation, though the effect appears to depend on the precise task being trained[42]. Importantly, activity increases were greatest for the anterior regions, both in the dorsal and ventral PFC, indicating a greater plasticity of these areas to task execution (Fig. 3). We should emphasize that differences between areas were quantitative rather than qualitative; a small percentage of the total variance was explained solely by a neuron's position along the AP axis. However, the effects were consistent, and neural variables beyond firing rate, including stimulus discriminability (Fig. 4), mutual information (Fig. 5), discharge variability (Fig. 7), and correlation between neurons (Fig. 7), all increased more in anterior than in posterior areas after training.

Perhaps the clearest demonstration of task-related activity was the difference between match and nonmatch responses, which increased markedly after training, when monkeys were required to perform this judgement. This task-related modulation of firing rate was greatest in the anterior areas (Fig. 6). Increased differentiation of whether a stimulus was a match or a nonmatch was expected, since the task specifically required monkeys to make a judgement on whether a stimulus was a match or a nonmatch and it is known that this difference is predictive of performance[33,34,43].

In principle, activity changes could be the effect of training-induced, long-term alterations in neuronal circuitry, causing new patterns to emerge and more so for anterior than posterior areas or may reflect plasticity at a much faster time scale, driven by factors related to the execution of the task[44]. Our experiment comparing responses during passive fixation and during execution of the task in the same neurons recorded after training revealed that both types of plasticity were present. However, differences between stages were quantitatively greater than differences between tasks. It is likely therefore that the long-term effects of training were the primary driving force behind the greater increases in activity in anterior areas.

Our analysis relied on parcellation of the prefrontal cortex into six regions, following loosely the organization proposed by Petrides and colleagues[14,45,46]. However this scheme is not entirely

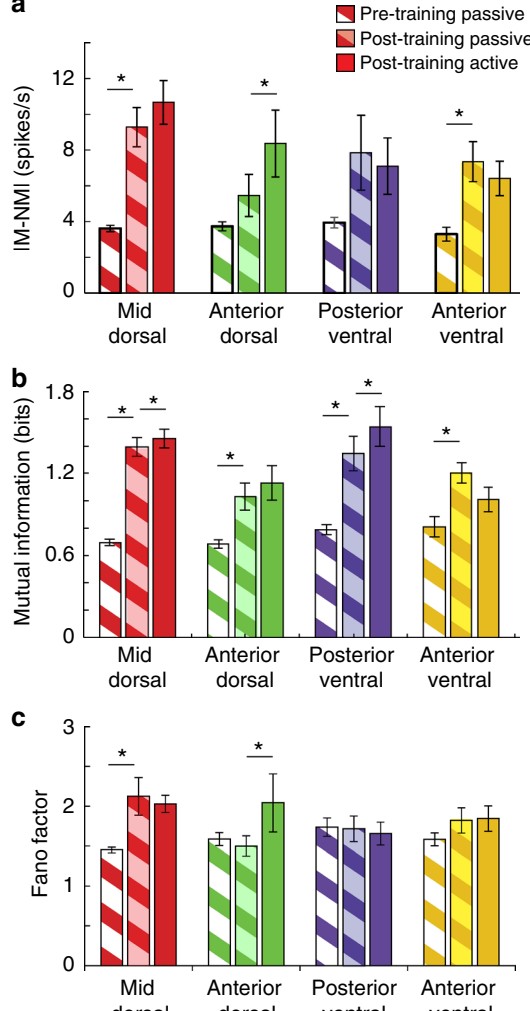

**Fig. 8** Task effects. **a** Comparison of match and nonmatch responses in neurons obtained prior to training, when all stimuli were presented passively, and after training, for neurons tested both with the passive fixation and with the active working memory task. Only neurons that exhibited significant elevation of firing rate for at least one stimulus condition, at one trial- epoch were included in this analysis (mid-dorsal: $n = 368$ pre-training, $n = 94$ post-training; anterior-dorsal: $n = 153$ pre-training, $n = 24$ post-training; posterior-ventral, $n = 178$ pre-training, $n = 22$ post-training; anterior-ventral: $n = 46$ pre-training, $n = 47$ post-training). Stars indicate significant differences in 1-tailed $t$-test between pre-training passive and post-training passive measures, or in 1-tailed, paired $t$-test in post-training passive vs. active comparisons. **b** Mean Mutual Information values in each prefrontal region. **c** Mean Fano factor values in each prefrontal region

agreed upon, and our parcellation only approximated the locations of cyto-architectonic borders. Treating location along the AP axis as continuous variable suggested a gradient of stimulus selectivity and task plasticity along the AP axis, though previous studies have identified abrupt transitions in some properties along the prefrontal surface[47]. Segmentation in additional prefrontal areas is also possible, as suggested by anatomical studies[45,48].

In the absence of neurophysiological evidence, until now, our understanding of the specialization of prefrontal subdivisions was informed by lesion studies, which have demonstrated dissociable effects on executing different cognitive tasks, depending on anatomical location. Thus, posterior-dorsal PFC lesions impair

performing tasks relying on associations (e.g., green light means press left button, red light means press right)[22]. Lesions of the mid-dorsal PFC result in deficits in tasks such as the delayed nonmatch to sample, requiring maintenance of a stimulus in memory[22]. Ventral prefrontal lesions do not produce impairments of performing tasks requiring recognition or simple recall; their effects only emerge in tasks requiring active retrieval of information from memory, such as free-recall tasks[49]. The ventral prefrontal cortex is also critical for learning a task by trial and error, and for reversal learning, which entails forming new associations during the course of a behavioral session[21,50]. Our recordings did not sample sufficiently the frontopolar region, but prior studies would predict little involvement in the representation of the stimuli or match/nonmatch information. Neurophysiological recordings reveal distinct patterns of activity in the frontopolar cortex than adjacent areas, including little selectivity for stimuli, but activation around the time of feedback and prior to self-generation of decisions[51,52]. Frontopolar lesions in monkeys impair one-trial learning of unfamiliar objects as well as learning of new task rules[53,54].

A hierarchical organization may imply that only the most anterior prefrontal areas influence behavior. Our Choice Probability analysis indicated that this was not the case; posterior areas exhibited no less influence on the output of behavior than anterior ones in agreement with previous studies[43]. The result suggests that much of the stimulus information necessary to perform the task is routed through posterior prefrontal areas, which influence the downstream determination of behavior, and that activation in one area cascades tightly through the prefrontal hierarchy. Ventral areas, too, which were not highly selective for the properties of the stimuli used in these experiments, exhibited a substantial increase in activity after training. Such activation may represent task rather than stimulus variables, without which correct execution may not be possible. Our experiments provide direct evidence that robust representation of stimulus information may emerge in the anterior prefrontal cortex following training if it is necessary for execution of a cognitive task. This is the essence of a hierarchy that routes information flexibly, depending on task demands.

## Methods

**Animals.** Six male rhesus monkeys (*Macaca mulatta*), age 5–9 yrs old, weighing 5–12 kg were used in these experiments. None of the animals had any prior experimentation experience at the onset of the experiments. Monkeys were either single-housed or pair-housed in communal rooms with sensory interactions with other monkeys. All experimental procedures followed guidelines by the U.S. Public Health Service Policy on Humane Care and Use of Laboratory Animals and the National Research Council's Guide for the Care and Use of Laboratory Animals and were reviewed and approved by the Wake Forest University Institutional Animal Care and Use Committee.

**Experimental setup.** Monkeys sat with their head fixed in a primate chair while viewing a monitor positioned 68 cm away from their eyes with dim ambient illumination. Animals were required to fixate on a 0.2° white square appearing in the center of the screen. During each trial, animals maintained fixation on the square while visual stimuli were presented either at a peripheral location or over the fovea in order to receive a liquid reward. Any break of fixation immediately terminated the trial and no reward was given. Eye position was monitored throughout the trial using a non-invasive, infra-red eye position scanning system (model RK-716; ISCAN, Burlington, MA). The system achieved a < 0.3° resolution around the center of vision. Eye position was sampled at 240 Hz, digitized and recorded. Visual stimuli display, monitoring of eye position, and the synchronization of stimuli with neurophysiological data were performed with in-house software[55] implemented on the MATLAB environment (Mathworks, Natick, MA), and utilizing the psychophysics toolbox[56].

**Pre-training presentation.** Following a brief period of fixation training and acclimation to the stimuli, monkeys were required to fixate on a center position while stimuli were displayed on the screen. The monkeys were rewarded for maintaining fixation during the trial with a liquid reward (fruit juice). The stimuli

shown were white 2° square stimulus presented in one of nine possible locations arranged in a 3 × 3 grid of 10° distance between adjacent stimuli. The same stimuli were shown following training in a working memory tasking during the "post-training" phase.

A fixation interval of 1 s where only the fixation point was displayed was followed by 500 ms of stimulus presentation, followed by a 1.5 s "delay" interval where, again, only the fixation point was displayed. A second stimulus was subsequently shown, either identical in location to the initial stimulus, or diametrically opposite the first stimulus. This second stimulus display was followed by another "delay" period of 1.5 s. The location and identity of stimuli in these experiments was of no behavioral relevance to the monkeys during the "pre-training" phase. In a few sessions, a variable delay period was used. Neurons recorded in these sessions appear in most analyses, though they are excluded from analyses that assume a fixed delay period (e.g., the PSTH of Fig. 3).

**Working memory task**. Four of the six monkeys were trained to complete spatial working memory tasks. The task used in most experiments (Fig. 1c) required the monkeys to remember the spatial location of the first stimulus shown, observe a second stimulus, and report whether the second stimulus was shown in the same location as the first stimulus or if it was in the diametrically opposite location via saccading to one of two target stimuli. For two of the monkeys, a match would mean a saccade to the green square stimulus while a nonmatch would mean a saccade to a blue square stimulus. Targets for the remaining monkey were an "H" and a diamond shape for match condition/nonmatch condition, respectively. Each target stimulus appeared at locations orthogonal to the cue/sample stimuli while the target feature locations were varied randomly from trial-to-trial. One of the four monkeys was trained in a different spatial task, a variant of the delayed response task[57]. Its structure was identical to the first five epochs of the match/nonmatch task except that the second stimulus always appeared in the same location as the first stimulus. After the end of the second delay period, the animal then had to saccade to the location where the stimulus was located. Neurons recorded from this animal are excluded from the match/nonmatch analysis.

**Surgery and neurophysiology**. A 20 mm diameter craniotomy was performed over the PFC and a recording cylinder was implanted over the site. The location of the cylinder was visualized with anatomical MRI imaging and stereotaxic coordinates post-surgery. For two of the four monkeys in the post-training phase (subjects MA and EL), the recording cylinder was moved after an initial round of recordings so that an additional surface of the prefrontal cortex could be sampled.

**Anatomical localization**. Each monkey underwent a magnetic resonance imaging scan prior to neurophysiological recordings. Electrode penetrations were mapped onto the cortical surface. We identified 6 lateral prefrontal regions: a posterior-dorsal region including area 8 A, a mid-dorsal region including area 8B and area 9/46, an anterior-dorsal region including area 9 and area 46, a posterior-ventral region including area 45, an anterior-ventral region including area 47/12, and a frontopolar region including area 10 (Fig. 1). The frontopolar region was not sampled sufficiently for this analysis and is depicted in Fig. 1 mainly to illustrate the boundaries of the anterior-dorsal and anterior-ventral regions.

In addition to comparisons between areas segmented in this fashion, other analyses were performed taking into account the position of each neuron along the AP axis. This was defined as the line connecting the genu of the arcuate sulcus to the frontal pole, for the purposes of this analysis. The recording coordinates of each neuron were projected onto this line and position was expressed as a proportion of the length of this line.

**Neuronal recordings**. Neural recordings were carried out in areas 8, 9, 9/46, 45, 46, and 47/12 of the PFC prior to training and following training in a spatial working memory task. Subsets of the data presented here were previously used to determine the properties of neurons in the dorsal and ventral prefrontal cortex pooled together[24], and properties of neurons prior to training[23]. Newly acquired data were added here, to determine differences before and after training in posterior-dorsal, mid-dorsal, anterior-dorsal, posterior-ventral, and anterior-ventral prefrontal subdivisions. Extracellular recordings were performed with multiple microelectrodes. These were either glass- or epoxylite-coated tungsten electrodes with a 250 μm diameter and 1–4 MΩ impedance at 1 kHz (Alpha-Omega Engineering, Nazareth, Israel). Arrays of up to 8-microelectrodes spaced 0.2–1.5 mm apart were advanced into the cortex with a Microdrive system (EPS drive, Alpha-Omega Engineering) through the dura into the prefrontal cortex. The signal from each electrode was amplified and band-pass filtered between 500 Hz and 8 kHz while being recorded with a modular data acquisition system (APM system, FHC, Bowdoin, ME). Waveforms that exceeded a user-defined threshold were sampled at 25 μs resolution, digitized, and stored for off-line analysis. Neurons were sampled in an unbiased fashion, collecting data from all units isolated from our electrodes, with no regard to response properties of a neuron being isolated. Recorded spike waveforms were sorted into separate units using an automated cluster analysis relying on the KlustaKwik algorithm[58], which applied principal component analysis of the waveforms. To ensure the stability of firing rate in the recordings analyzed, we identified recordings in which a significant effect of trial sequence was

evident on the baseline firing rate (ANOVA, $p < 0.05$), e.g., due to a neuron disappearing or appearing during a run, as we were collecting data from multiple electrodes. Data from these sessions were truncated so that analysis was only performed on a range of trials with stable firing rate. Less than 10% of neuronal records were corrected in this way.

Identical data collection procedures, recording equipment, and spike sorting algorithms were used before and after training, to ensure that any changes reported between stages were not due to these factors. To also ensure that changes in neuronal firing properties were not the result of systematic differences in the inherent properties of neurons sampled, we compared the Signal-to-Noise Ratio (SNR) of neuronal recordings before and after training[10]. For each unit, we defined SNR as the ratio of the peak-to-trough height of its mean action potential waveform, divided by the standard deviation of the noise. The latter was computed from the baseline of each waveform, derived from the first 10 data points (corresponding to 0.25 ms) of each sample. SNR provides an overall measure of unit isolation quality[10]. We used this measure to identify neurons with excellent isolation, defined based on SNR > 5.

**Data analysis**. All data analysis was implemented with the MATLAB computational environment (Mathworks, Natick, MA). Neuron analysis involved, initially, determining the mean firing rate of each neuron in each trial epoch: 1 s of fixation; 0.5 s of cue presentation; 1.5 s of first delay period; 0.5 s of sample presentation; and 1.5 s of second delay period. Next, we compared responses of each neuron in the 1 s baseline, fixation period with the cue presentation period and the delay period following it. Any neuron that had significantly greater responses during the cue or delay was identified as a task-responsive neuron (one-tailed $t$-test; $p < 0.05$). Neuronal selectivity for the stimuli locations was evaluated by using a selectivity index defined as (Max − Min)/(Max + Min), where Max is the maximum response (during the stimulus presentation period, averaged across trials) and Min is the response to the diametrically opposite stimulus. The selectivity index values were then averaged across neurons of a region, and means were compared between regions. ROC analysis was used to compare the distribution of firing rates of a neuron for two stimulus conditions: that involving presentation of the stimulus at the neuron's best and worst location. The area under the ROC curve represents the probability an ideal observer can discriminate between the two stimulus conditions, based on the relative difference in firing rate distributions between the two stimulus conditions[27]. The analysis was performed in a time-resolved fashion, in sliding 250 ms bins, stepped every 20 ms. ROC values were calculated for neurons, as a function of their position along the AP axis. Mean values were binned in overlapping windows extending by a length equal to 4% of the total length of the axis and stepped every 1% of the axis. Then values were convolved with a Gaussian curve of standard deviation equal to 2% of the length of the axis. ROC analysis was also performed to compare correct and error trials, following cue appearance at the same location (the best for each neuron). The area under this ROC curve is known as Choice Probability[35]. A Mutual Information statistic was calculated to determine how well stimulus locations can be discriminated from each other, based on the spike counts of a neuron[27,28,59]. Mutual information was calculated in the entire delay period and values were compared between areas and training phases. The Fano factor of a neuron's discharge rate (defined as the variance divided by the mean) was estimated in the stimulus presentation and delay periods, as described previously[60]. Data for each neuron and stimulus location were initially treated separately. Spike counts were computed in a 100-ms sliding window moving in 20-ms steps. We computed the variance and mean of the spike count across trials and performed a regression of the variance to the mean. This slope of this regression represents the Fano Factor reported here. Spike-count correlation was determined from pairs of neurons recorded simultaneously from separate electrode spaced 0.5–1.0 mm apart, as described previously[61]. Recordings of neuron pairs selected for analysis were required to contain at least 1000 spikes in total, and no less than 100 spikes in each neuron, to avoid artificially low noise correlation values for neurons with low firing rate, when inadequate numbers of trials are available[29]. For each pair, we used the position that corresponded to the mean of the two neurons' projection on the AP axis. The mean and standard deviation of firing rate across trials was calculated for each stimulus presentation and following delay period. The firing rate of each trial had the mean firing rate of the corresponding stimulus condition subtracted from it, and then it was divided by the standard deviation to provide a normalized estimate independent of stimulus. The Pearson correlation coefficient between these normalized firing rate values provides the spike count correlation. The absolute difference between firing rate elicited by the stimulus appearing at the best location of each neuron when it appeared as a match, and at the same location when it appeared as a nonmatch was used to determine the neuron's match/nonmatch preference. For the same comparison, d′ was also computed, defined as $|\mu_M - \mu_{NM}|/\sqrt{[(\sigma_M^2 + \sigma_{NM}^2)/2]}$, where $\mu_M$ and $\sigma_M$ denote the mean and standard deviation of the match rates, and $\mu_{NM}$ and $\sigma_{NM}$ the mean and standard deviation of the nonmatch rates, respectively.

## Data availability

All relevant data and code will be available from the corresponding author on reasonable request.

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

## Acknowledgements

Research reported in this paper was supported by the National Eye Institute of the National Institutes of Health under award numbers R01 EY017077 and R01 EY016773 to C.C.; by NINDS training grant T32 NS073553; NIMH fellowship F31 MH104012 to M.R.R; and by the Wake Forest Clinical and Translational Science Institute. We wish to thank David T. Blake for helpful comments on an earlier version of this manuscript, Fumi Katsuki for computational contributions and Kathini Palaninathan and George Pate for technical help.

## Author contributions

Conceptualization: C.C. Neurophysiological recordings: M.R.R. and X.L.Q. Data analysis: M.R.R., X.L.Q., and X.Z., C.C. Manuscript writing: M.R.R. and C.C. Review and editing: all authors.

## Additional information

**Competing interests:** The authors declare no competing interests.

