## [Peer Review File · Nature Communications]

Reviewers' comments:

Reviewer #1 (Remarks to the Author):

This is an interesting and well-written paper that contributes to our understanding of signaling in anterior to posterior prefrontal cortical divisions associated with the performance of cognitive tasks.

Comments

1. This study addresses the question of the nature of functional specialization within different anterior and posterior (and dorsal and ventral) regions of the primate prefrontal cortex. In the Introduction, the authors review existing literature that provides strong evidence for anterior-posterior specialization (obtained from numerous anatomical, imaging, lesion and neurophysiological studies) and briefly touch on studies that provide modest evidence or do not support this idea. [For the latter, they state: "Yet, experiments in animals trained to perform cognitive tasks have largely failed to account for such specialization, with some reporting no obvious specialization between dorsal and ventral areas and others documenting only modest differences between regions along the anterior posterior axis of the prefrontal cortex."]. Thus the authors acknowledge that the preponderance of published evidence supports the notion that there is a gradient of functional regions across the anterior-posterior axis of the prefrontal cortex. Within this context of cited existing evidence for functional specialization in regions of the primate prefrontal cortex the novelty and impact of the study thus seems incremental. In other words, the development of a stronger argument for the novelty and importance of their study would make the paper more compelling. A suggestion- perhaps emphasize further that the really novel aspect of the study is the pre- versus post- training study, because direct evidence of learning-induced changes in task-related neuronal firing in primate PFC is scarce and it is largely unknown if learning affects neuronal firing in differentially across distinct areas of the prefrontal cortex.

2. A large number of neurons were studied in this project, which is a positive. However, an examination of the scatter plots in figures 3-5 raise questions of the appropriateness of the use of linear fits to the data. For example, the authors state that "Prior to training, firing rate did not vary appreciably as a function of dorsal anterior-posterior location (Fig. 3D), as evidenced by a slope of a regression line that was not significantly higher than zero ($F_{1,1206}=0.03$, $p=0.872$). On the contrary, after training (Fig. 3E) the slope of the regression line was highly significant ($F_{1,1082}=17.83$, $p=2.6 \times 10^{-5}$), as was the increase of the post-training slope relative to the pre-training one (F-test, $F_{1,2288}=15.31$, $p=0.0001$)."

Examination of several of the scatter plots reveal poor correlation of firing rates with AP location and leads to the question of whether significance may be driven by the high degrees of freedom. It would be helpful if the authors would provide R values as indicators of the strength of the relationships. Furthermore, rather than using scatter plots of each individual neuron's firing rate across the AP location, the authors should (also) plot the mean firing rate of all neurons in a given area for each subject and fit those data.

3. Mean data should be shown as vertical dot plots (wherein the mean for each subject is represented by one point) rather than as bar graphs.

Reviewer #2 (Remarks to the Author):

In the paper, the authors describe the properties of neurons across multiple areas of nonhuman primate prefrontal cortex, including area 8A, 46, 9, 45, 47 and 10 while animals either fixated and passively viewed visual stimuli or during the learning and execution of a spatial working memory task. A large sample (>3000) of neurons were recorded from across the prefrontal areas in 4 monkeys. The authors report that the effects of training, which include an increase in firing rate during the working memory delay period, were more pronounced in anterior prefrontal cortex.

The dataset studied in the paper is truly impressive, in terms of the number of neurons studied and the anatomical coverage, well as the testing prior to and after training on the working memory task. This aspect of the study is the true highlight. Unfortunately, what is missing in the study is a more detailed (some might say 'adequate') analysis of the data presented, and this issue greatly limits the apparent significance of the conclusions that can be drawn. I discuss this in more detail below.

1. The reliance upon non-normalized firing rates and firing rate changes (figures 2-3, 5) in the analysis makes it difficult to place too much stock in the observations given the usual caveat that differences in the statistics of firing rates across neurons will result in high-firing-rate neurons having more impact on the group averages. There is almost never a good reason not to normalize the spike rates (or changes thereof) in population averages. Since the comparisons across regions and across training phases are between comparisons, it is always possible that the differences in firing rate observed are due to, or obscured by, differences in recording quality, e.g. ability to isolate neurons that occur across different anatomical locations, or across recording sessions. Although simply normalizing the firing rates of each neuron will only mitigate this issue, i.e. by reducing the influence of differences in the distributions of firing rate (across area and across training epochs) on the population average, it will help nonetheless.

2. The use of the ROC analysis in figure 4 is a good step toward a more rigorous analysis of the dependence of task-relevant signaling on the anatomical location, but it still falls short of the what seems possible with the dataset. The authors used more than just 2 locations ('best' and 'worst'), thus one would preferable like to know if information about all of those locations is more available from the responses of anterior dorsal neurons. So perhaps using a mutual information measure (e.g. Bushman and Miller, 2007, Science) would better as it allows the authors to deploy all of their data instead of throwing out much of it.

3. Lastly, in many cases, recordings were made with multiple electrodes, and thus not only can the authors look at how pseudo populations of neurons from different areas encode different task variables across training epochs using data across recordings, but they can

also potential look at the activity of (small) populations of neurons recorded simultaneously across different structures.

4. I think that perhaps using 1 or all of the above suggestions the authors might uncover some observations that lift the significance of the results reported in the paper thus far.

Reviewer #3 (Remarks to the Author):

The PFC can be divided into different subregions. Their response patterns and functions are not well understood, especially in the context of learning. It is an important but difficult question in the field. Riley et al. carried out an interesting and ambitious study to look at how the response patterns of different PFC subregions are changed after monkeys learning a working memory task. The authors concluded that a difference existed along the anterior-posterior axis. The more anterior part of the PFC appeared to be more plastic.

However, the data and the analyses have many problems that may affect the conclusion. In addition, the analyses and figures were not adequately described to allow the readers to assess the validity of the claims.

The biggest weakness of the manuscript is a lack of efforts to exclude alternative explanations. Claiming there is a difference in response patterns due to the training requires one to carefully control the other factors. The current dataset unfortunately has several confounders that are not sufficiently addressed or discussed in the manuscript.

1. The biggest confounder is the task effect. Unless I misunderstood the paper, in which case the authors should've made their writing clearer, the pre-training data came from recordings during the passive-viewing task, while the post-training data came from the working memory task. Thus, the reported training effects could be just task effects.
2. The second confounder is the recording locations. Were the neurons sampled in each subarea sufficiently representative? Based on Fig S1, the recordings only covered part of each subarea. I know it is hard to cover a subarea completely. But the authors should at least show evidence that the coverage of each subarea was consistent before and after the recording. This is not clear based on Fig S1.
3. Lastly, one needs to consider whether the effect may be due to individual monkey difference. Many pre-/post-training data came from different monkeys. This is a problem especially for the anterior part of the brain, where data basically came from only 1.5 monkeys (NI pre/post training and MA pre-training only, based on Fig S1).

Other issues:

1. Fig 2: What's time 0 on the x-axis? Is it the fixation-on? I don't see an increased discharge during the delay period based on this figure. Do I miss something? Any statistics?
2. Fig 3D-K: I'd like to see neurons from different monkeys labeled.
3. Fig 4E: What period are the ROC values from? In addition, the calculation of ROC is problematic. The authors claim that the ROC is calculated with the best and the worse

stimulus location. It is unclear what's the statistic test used to define the best and the worst. And then, if the authors used some statistic test to figure out the best and the worst with some p-value level, the ROC should be also significantly higher than 0.5. It's a bit circular.

4. Fig 5: What period are the firing rates calculated from?

5. Many analyses are pooling neurons from both dorsal and ventral regions together and look at only their AP locations (Fig 3D-K, Fig 4D-E, Fig 5B-E). This is questionable. There are clear differences between the dorsal and ventral areas. Their APs cannot be simply aligned. I'd like to see them separated.

We wish to thank the reviewers for their careful reading of our manuscript and constructive comments. We have extended the study considerably, performed new analyses, and revised the text to address the issues raised in review (in italics). As a result, we believe the manuscript has been strengthened considerably. We have additionally made changes to the text, according to Nature Communications guidelines: abstract reduced to 150 words, main text within 5,000 words (actually slightly exceeded, based on review recommendations), discussion headings removed, number of references <70.

Reviewer #1:

1. In the Introduction, the authors review existing literature that provides strong evidence for anterior-posterior specialization (obtained from numerous anatomical, imaging, lesion and neurophysiological studies) and briefly touch on studies that provide modest evidence or do not support this idea. [...] Thus the authors acknowledge that the preponderance of published evidence supports the notion that there is a gradient of functional regions across the anterior-posterior axis of the prefrontal cortex. Within this context of cited existing evidence for functional specialization in regions of the primate prefrontal cortex the novelty and impact of the study thus seems incremental. In other words, the development of a stronger argument for the novelty and importance of their study would make the paper more compelling. A suggestion- perhaps emphasize further that the really novel aspect of the study is the pre- versus post- training study, because direct evidence of learning-induced changes in task-related neuronal firing in primate PFC is scarce and it is largely unknown if learning affects neuronal firing in differentially across distinct areas of the prefrontal cortex.

Response: The reviewer's point is well taken. We did not wish to ignore prior studies in order to inflate the significance of the anterior-posterior specialization idea, and we cited upfront the evidence in favor of specialization between prefrontal areas, as the reviewer acknowledges. That being said, we now emphasize the truly novel aspect of our study which is that learning effects neuronal firing differentially across distinct areas of the prefrontal cortex – precisely as the reviewer points out.

2. A large number of neurons were studied in this project, which is a positive. However, an examination of the scatter plots in figures 3-5 raise questions of the appropriateness of the use of linear fits to the data. Examination of several of the scatter plots reveal poor correlation of firing rates with AP location and leads to the question of whether significance may be driven by the high degrees of freedom. It would be helpful if the authors would provide R values as indicators of the strength of the relationships. Furthermore, rather than using scatter plots of each individual neuron's firing rate across the AP location, the authors should (also) plot the mean firing rate of all neurons in a given area for each subject and fit those data.

Response: These were excellent suggestions. We now provide R values in the text for all regression analyses. We acknowledge that differences between areas are quantitative rather than qualitative, and that position along the AP axis by itself explains only a limited fraction of the variance of the variables under study. We now plot averages based on individual subject means for Figures 3-5. Even though unequal sample sizes were obtained from different monkeys, these regression lines provided remarkably similar with the regression lines based on individual neurons pooled across monkeys. This result provides evidence that the results are consistent across animals, and not driven by one or two individuals.

3. Mean data should be shown as vertical dot plots (wherein the mean for each subject is represented by one point) rather than as bar graphs.

Response: We have now added-single subject averages plotted over the bar mean. As noted above, sample sizes between subjects varied considerably thus a handful of neurons from a single subject might have little influence on the overall mean. By combining the two, we believe we have provided a more complete picture of our results.

Reviewer #2:

What is missing in the study is a more detailed (some might say ‘adequate’) analysis of the data presented, and this issue greatly limits the apparent significance of the conclusions that can be drawn. I discuss this in more detail below.

Response: The reviewer’s point is well taken. We have now extended the study considerably – even beyond space limitations.

1. The reliance upon non-normalized firing rates and firing rate changes (figures 2-3, 5) in the analysis makes it difficult to place too much stock in the observations given the usual caveat that differences in the statistics of firing rates across neurons will result in high-firing-rate neurons having more impact on the group averages.

Response: We do see the point and we would have liked to do exactly what the reviewer is describing – however, we sampled different sets of neurons before and after training, from different electrode tracks. There is no simple way of matching neurons recorded in the two stages of the experiment with each other, which would allow us to use a common, normalized firing rate. If we simply normalize the firing rate of each neuron with itself, then all averages of Figures 2-3 would simply scale to 1 (we did try it, and it was not informative). It is precisely for this reason that we relied on the Selectivity Index in Fig. 3L-S, which essentially normalizes firing rate, and weighs low firing neurons no less than high firing ones. We clarify that in the text now.

2. The use of the ROC analysis in figure 4 is a good step toward a more rigorous analysis of the dependence of task-relevant signaling on the anatomical location, but it still falls short of the what seems possible with the dataset. The authors used more than just 2 locations (‘best’ and ‘worst’), thus one would preferable like to know if information about all of those locations is more available from the responses of anterior dorsal neurons. So perhaps using a mutual information measure (e.g. Bushman and Miller, 2007, Science) would better as it allows the authors to deploy all of their data instead of throwing out much of it.

Response: This was an excellent suggestion. We now include Mutual Information analysis in a new section of the paper, and a new Figure 5.

3. Lastly, in many cases, recordings were made with multiple electrodes, and thus not only can the authors look at how pseudo populations of neurons from different areas encode different task variables across training epochs using data across recordings, but they can also potential look at the activity of (small) populations of neurons recorded simultaneously across different structures.

Response: This was also a great suggestion but unfortunately the size of our typical ensembles of 5-6 neurons was not large enough to perform this decoding analysis, particularly since only a minority of neurons was informative about the stimuli (see Fig. 3A). We have taken one step towards this direction by examining the correlated firing of **pairs** of neurons, which in fact demonstrates that spike-count correlation changes are greater in anterior areas after training (Fig. 6).

4. I think that perhaps using 1 or all of the above suggestions the authors might uncover some observations that lift the significance of the results reported in the paper thus far.

Response: We appreciated the reviewer’s constructive comments. In addition to the analyses suggested, we have now included analysis of neuronal variability, quantified as the Fano factor of spike counts (Fig. 6A-D), which also demonstrated increased effects after training in the anterior areas. We believe the addition of all these analyses provides a much more complete picture, now.

Reviewer #3:

1. The biggest confounder is the task effect. Unless I misunderstood the paper, in which case the authors should've made their writing clearer, the pre-training data came from recordings during the passive-viewing task, while the post-training data came from the working memory task. Thus, the reported training effects could be just task effects.

Response: The reviewer raises an important point, which apparently we failed to make clear in our original submission. To address the effects of task, we compare recordings obtained before and after training while the monkeys were viewing stimuli passively and recordings after training when the monkey viewed stimuli passively or performed the task. These comparisons allow us to quantify the effects of training and task execution. We have now extended this analysis considerably, and dedicate a separate figure (Fig. 8) and a section of the paper titled "Effects of training and of task execution", for added clarity.

2. The second confounder is the recording locations. Were the neurons sampled in each subarea sufficiently representative? Based on Fig S1, the recordings only covered part of each subarea. I know it is hard to cover a subarea completely. But the authors should at least show evidence that the coverage of each subarea was consistent before and after the recording. This is not clear based on Fig S1.

Response: This is also an important point. We address that in two ways. By plotting responses of each neuron as a function of Anterior-Posterior location and treating location as a continuous variable, we are able to make robust comparisons about the variables in question before and after training. Additionally, we now plot averages based on individual monkeys (Fig. 3 and Fig. 7). The results largely verify the findings based on individual neurons pooled across monkeys.

3. Lastly, one needs to consider whether the effect may be due to individual monkey difference. Many pre-/post-training data came from different monkeys. This is a problem especially for the anterior part of the brain, where data basically came from only 1.5 monkeys (NI pre/post training and MA pre-training only, based on Fig S1).

Response: We now report results separately for individual monkeys. Data from separate monkeys tend to fall on the same regression line (Fig. 3, 7 and supplementary information) providing evidence that the effects we document are consistent across individuals. Importantly, we now emphasize that the effects of training we reported in the anterior regions were in agreement with previous reports in the literature that have described robust modulation of prefrontal neurons by task variables, which presumably appear after training. It was the relative **lack of changes in the posterior areas** after training (which were sampled densely, in multiple monkeys) that were responsible for the gradient of plasticity that we observed across the anterior-posterior axis of the prefrontal cortex (Fig. 3-7). We should also point out that neurophysiological studies of the prefrontal cortex rarely rely on more than 2 animals, and what has been established in the literature about the properties of neurons in different areas from prior studies has almost invariably been derived from 1-2 individuals. We believe our study is a significant addition in the literature, in this respect.

Other issues:

1. Fig 2: What's time 0 on the x-axis? Is it the fixation-on? I don't see an increased discharge during the delay period based on this figure. Do I miss something? Any statistics?

Response: We now clarify that time 0 is indeed the onset of fixation. The reviewer is correct that the increase in firing rate during the delay period is subtle, particularly prior to training, when the monkey is not doing a working memory task. Only a minority of neurons exhibit sustained increase in firing rate during the delay, even after training, whereas some neurons are inhibited during the delay period. We also explain that Fig. 2 depicts responses that were selected based on the best *cue* (1st stimulus) response of each neuron. We plot the percentages of neurons that exhibit significant increase of firing rate in the delay period over the fixation period (evaluated with paired t-tests) in Fig. 3A.

2. Fig 3D-K: I'd like to see neurons from different monkeys labeled.

Response: We now plot means from individual monkeys, as reviewer #1 also requested.

3. Fig 4E: What period are the ROC values from? In addition, the calculation of ROC is problematic. The authors claim that the ROC is calculated with the best and the worse stimulus location. It is unclear what's the statistic test used to define the best and the worst. And then, if the authors used some statistic test to figure out the best and the worst with some p-value level, the ROC should be also significantly higher than 0.5. It's a bit circular.

Response: These are important points that apparently we failed to make clear in our initial manuscript. ROC analysis compares the distribution of firing rates generated by two different stimulus conditions. We compared the conditions that generated the highest and lowest firing rates during the stimulus presentation period. These, indeed, would be expected to be higher than 0.5 by definition, and the fact that they do is of no consequence. In Fig. 4D-E we show results from the delay period and we are using this analysis to quantify **how large** this difference is in each area, before and after training. We, in fact, find systematic differences between areas and conditions. The result is telling in that the firing rate reliability differs between areas. In addition, we now present results from Mutual Information analysis, which determines how well all nine locations can be differentiated from each other, without the need to identify the best and worst response (Fig. 6).

4. Fig 5: What period are the firing rates calculated from?

Response: We clarify that this is the sample (second stimulus) presentation period (now Figure 7). This is critical for the interpretation of these results. Here we assess the task effects that can only be present once both stimuli have been displayed, and the monkey can make a judgment on whether they are the same or not, in the context of the task.

5. Many analyses are pooling neurons from both dorsal and ventral regions together and look at only their AP locations (Fig 3D-K, Fig 4D-E, Fig 5B-E). This is questionable. There are clear differences between the dorsal and ventral areas. Their APs cannot be simply aligned. I'd like to see them separated.

Response: This is also a critical issue. We did not plot these data together in any panel in the original manuscript but we used different symbols to display dorsal and ventral neurons. We now added labels to every figure panel, indicating the area, for added clarity.

Reviewers' comments:

Reviewer #1 (Remarks to the Author):

The authors have responded comprehensively to the comments and suggestions of the reviewers and this has led to a substantially improved and more compelling paper. The study is expertly done and the results are important and will be of significant interest to the neuroscience community.

Reviewer #2 (Remarks to the Author):

The authors have addressed the majority of my concerns and I think the paper has improved notably. They also seem to have addressed most of the important concerns of the other reviewers. The significance of the (large amount of) data presented is now much easier to appreciate.

Reviewer #3 (Remarks to the Author):

I am thankful for the efforts the authors made to address many of the concerns in the original manuscript. While this manuscript is improved, there are still some serious issues that remain to be resolved.

Major issues:

1. While the new Fig 8 is great, it's still unclear to me whether the post-training data in the other figures were from the active or the passive task. It would only be appropriate if all comparisons were between recordings from the same passive viewing task in the pre- and post-training periods. The authors need to be crystal clear about this in the writing.
2. In Figure 2, there's apparently a change in the baseline activity in all panels. I'm afraid this might be an indication of artifacts. Suspects include using different electrodes or recording equipments, different sorting methods, or maybe monkeys getting more mature. If training does change the baseline activity, I'd like to see some discussion.
3. How do the spike correlation analyses support the claim that the anterior regions are more affected by cognitive factors (Fig 6E-H)? If neurons have higher correlations, there's more redundancy in whatever they are encoding, right? Not sure how this can be translated into the conclusion authors made.
4. Fig 7 is flawed without considering the firing rate variance. Especially we've seen that the baseline firing is different (Fig 2). It would be much better if something like d' is used instead to quantify how well neurons discriminate match vs. nonmatch trials.
5. The "Choice Probability" analyses are not really choice probability analyses, as first described by Britten et al. It's still interesting to look at how neurons behave in correct and error trials, but this is not how choice probability is defined in the field. The authors should give the analyses a different name to avoid confusion.
6. Fig 8A looks very different from Fig 7A. My understanding is that, for each area, two of the 3 bars in Fig 8A should be identical with the two bars in Fig 7A. Am I missing

something?

7. I have my reservations about the overclaiming in the writing. While this is a very difficult study that has generated lots of interesting data, it is about a very simple spatial working memory task. The conclusions the authors made may be true for spatial memory tasks, but entirely different things might be found for other kinds of experiments. If one trains animals with an auditory discrimination task and observes plasticity in the auditory cortex, you cannot make the conclusion the auditory cortex is more plastic than the visual cortex. I hope the authors may acknowledge this point and scale back their claims.

Minor issues:

Line 40: extra "and" in the end of the line.

Line 44 "Our results reveal...": You cannot make that conclusion given the most anterior part of the brain (polar) is not studied in this study. And "necessary" is not supported by the current study.

Line 176-178 "A virtually identical...": The statement is a confusing. Figures 3M and 3Q look different, and I don't see an increase of slopes. Maybe authors mean less negative?

Line 483-497: The discussion of "choice probability" makes no sense. Proper choice probability gives you information how neurons firing might be read out in the downstream areas to generate choice. Here, the prefrontal cortex IS the downstream area. The authors' definition of choice probability looked at the difference of correct vs. wrong, which by themselves are results of decision making.

Multiple places: Please report R2 instead of R in the statistics. This is more conventional.

Reviewer #1:

The authors have responded comprehensively to the comments and suggestions of the reviewers and this has led to a substantially improved and more compelling paper. The study is expertly done and the results are important and will be of significant interest to the neuroscience community.

Reviewer #2:

The authors have addressed the majority of my concerns and I think the paper has improved notably. They also seem to have addressed most of the important concerns of the other reviewers. The significance of the (large amount of) data presented is now much easier to appreciate.

Response: We wish to thank the reviewers for their positive evaluation and acknowledgement of the significance of our study. We are grateful for their prior comments.

Reviewer #3:

I am thankful for the efforts the authors made to address many of the concerns in the original manuscript. While this manuscript is improved, there are still some serious issues that remain to be resolved.

Response: We do appreciate the reviewer's comments, which have helped us make our paper clearer and more accessible for a wider audience, and at the same time more thorough and including more technical detail for the aficionados in the field.

1. While the new Fig 8 is great, it's still unclear to me whether the post-training data in the other figures were from the active or the passive task. It would only be appropriate if all comparisons were between recordings from the same passive viewing task in the pre- and post-training periods. The authors need to be crystal clear about this in the writing.

Response: We believed we made this point clear since the original version of the manuscript. We now state in every section of the paper that we first report responses during task execution, which mirrors almost the entirety of the published prefrontal neurophysiology literature. We compare this dataset with the data collected during passive presentation of stimuli prior to training. Then, the section text titled "Effects of training and of task execution" we document precisely the comparison the reviewer refers to. We respectfully disagree that it would only be appropriate if all comparisons were between recordings from the same passive viewing task in the pre- and post-training. Had we not collected data during the active execution of the task, we would have missed what we believe is a major finding of our study: that the cumulative effect of training **and** task execution makes little difference on posterior prefrontal responses. Implicit in the reviewer's comment is the assumption that the representation of stimuli is entirely plastic according to task demands. We show that this is not necessarily the case for the posterior areas. We believe our analysis of both active and passive responses after training provides the complete picture.

2. In Figure 2, there's apparently a change in the baseline activity in all panels. I'm afraid this might be an indication of artifacts. Suspects include using different electrodes or recording equipments, different sorting methods, or maybe monkeys getting more mature. If training does change the baseline activity, I'd like to see some discussion.

Response: This is an important point that we recognize we did not address fully. We now point out that identical recording techniques, data acquisition equipment, and data sorting algorithms were used in the two recording stages. We include in more detail in the Methods the measures taken to ensure consistent quality control. We also add in the manuscript the control analyses based on a subset of recordings with excellent spike isolation obtained before and after training, identified based on SNR of spike waveform >5

(Supplementary Figure S3). Finally, we report in the text the ages of monkeys, for each recording stage and area (see also Supplementary Table 2). The animals were adult at the time the pre-training recordings were obtained; and recordings were obtained from each area at virtually identical ages.

After ruling out all of these potential sources of differences between conditions, we report that changes in baseline were in fact evident, which we discuss in more detail now (see also Supplementary Figure 2). Importantly, this shifts in baseline differed systematically between areas, which also argues against the effect of non-training-driven differences in recordings before and after training.

3. How do the spike correlation analyses support the claim that the anterior regions are more affected by cognitive factors (Fig 6E-H)? If neurons have higher correlations, there's more redundancy in whatever they are encoding, right? Not sure how this can be translated into the conclusion authors made.

Response: This is a subtle point that we recognize we did not explain sufficiently for non-specialists. Higher correlations between neurons are indeed suggestive of redundancy in terms of decoding the stimulus information. The critical issue, however, is the source of these correlations. We now explain that theoretical and experimental findings suggest that these are primarily the effects of shifts in attention (or motivation, or other cognitive factors) which affect jointly the neurons under study, causing their firing rate to shift in unison, from trial to trial. Spike count correlation changes are greater in areas where the effect of these factors is stronger. We cite these recent articles:

Denfield, G.H., Ecker, A.S., Shinn, T.J., Bethge, M. & Tolias, A.S. Attentional fluctuations induce shared variability in macaque primary visual cortex. *Nature communications* 9, 2654 (2018).

Ecker, A.S., Denfield, G.H., Bethge, M. & Tolias, A.S. On the Structure of Neuronal Population Activity under Fluctuations in Attentional State. *J Neurosci* 36, 1775-1789 (2016).

4. Fig 7 is flawed without considering the firing rate variance. Especially we've seen that the baseline firing is different (Fig 2). It would be much better if something like d' is used instead to quantify how well neurons discriminate match vs. nonmatch trials.

Response: What is being plotted in Figure 7 is the difference between mean rates of two conditions (match, nonmatch) recorded from each neuron, which is not affected by the neuron's baseline firing rate. Nonetheless, we now add analysis expressed in d' units in a new figure panel (Fig. 7B). This analysis produced the same pattern of changes as that of the raw firing rate difference.

5. The "Choice Probability" analyses are not really choice probability analyses, as first described by Britten et al. It's still interesting to look at how neurons behave in correct and error trials, but this is not how choice probability is defined in the field. The authors should give the analyses a different name to avoid confusion.

Response: We use the term Choice Probability as it has been used extensively in the literature to describe responses of prefrontal neurons. See for example:

Zaksas, D. & Pasternak, T. Directional signals in the prefrontal cortex and in area MT during a working memory for visual motion task. *J Neurosci* 26, 11726-11742 (2006).

Mendoza-Halliday D, Torres S, Martinez-Trujillo JC. Sharp emergence of feature-selective sustained activity along the dorsal visual pathway. *Nature Neurosci.* 2014 Sep;17(9):1255-62

Nonetheless, we now explain in more words exactly what we refer to in this analysis: "The area under the ROC curve comparing the distribution of firing rates in correct and error trials involving the same stimulus and only differing in terms of the monkey's choice".

6. Fig 8A looks very different from Fig 7A. My understanding is that, for each area, two of the 3 bars in Fig 8A should be identical with the two bars in Fig 7A. Am I missing something?

Response: This was also explained in the text, but we make clearer now. After training, it was not possible to hold isolation and record from all neurons both in the active and in the passive condition. Data for the passive-presentation experiment were acquired from a limited set of sessions, and these tended to be more often

sessions in which when neurons responsive to the task were encountered (because neurons not responding to any stimulus in our set were less likely to be informative about whether passive presentation or active task execution affects their responses). For this reason, we felt that the most conservative approach for comparing passive-presentation data acquired before and after training would be to limit our comparison to neurons that responded during the trial, identified with identical selection criteria both before and after training. Using all available pre-training neurons (from figure 7) only exaggerates the difference between the pre- and post-training passive presentation that we reported in the text. We have added these data now in an additional Supplementary Figure (S10), in which the pre-training match/nonmatch data are indeed identical to the bars shown in Figure 7A.

7. I have my reservations about the overclaiming in the writing. While this is a very difficult study that has generated lots of interesting data, it is about a very simple spatial working memory task. The conclusions the authors made may be true for spatial memory tasks, but entirely different things might be found for other kinds of experiments. If one trains animals with an auditory discrimination task and observes plasticity in the auditory cortex, you cannot make the conclusion the auditory cortex is more plastic than the visual cortex. I hope the authors may acknowledge this point and scale back their claims.

Response: The reviewer's point is well taken. We now qualify our conclusions in the abstract (see also response to minor point below), and we acknowledge the caveat that we don't know if the effect generalizes across tasks, in the first paragraph of the discussion, and onward.

Minor issues: Line 40: extra "and" in the end of the line.

Response: Corrected.

Line 44 "Our results reveal...": You cannot make that conclusion given the most anterior part of the brain (polar) is not studied in this study. And "necessary" is not supported by the current study.

Response: We have revised the abstract to refer to the "anterior aspects of the lateral prefrontal cortex" and replaced the clause suggesting that this is necessary for flexible behavior with "possess greater plasticity based on task demands".

Line 176-178 "A virtually identical...": The statement is a confusing. Figures 3M and 3Q look different, and I don't see an increase of slopes. Maybe authors mean less negative?

Response: Yes, that is what we meant. We have rephrased.

Line 483-497: The discussion of "choice probability" makes no sense. Proper choice probability gives you information how neurons firing might be read out in the downstream areas to generate choice. Here, the prefrontal cortex IS the downstream area. The authors' definition of choice probability looked at the difference of correct vs. wrong, which by themselves are results of decision making.

As noted in point #5 before, there is extensive precedent for this analysis in the prefrontal cortex. Our point is that a hierarchy of areas exists even within the prefrontal cortex, so that anterior areas are more downstream to posterior ones. Nonetheless, for added clarity, we have revised this paragraph to refer to "ROC values comparing correct and error trials".

Multiple places: Please report R² instead of R in the statistics. This is more conventional.

Response: Reviewer #1 requested that we report R values in the previous round of review and we complied. Correlation coefficients are no less frequently reported than coefficients of determination (R²) in similar studies. To avoid any confusion, we now report these values with lower case "r" rather than upper case "R".

REVIEWERS' COMMENTS:

Reviewer #3 (Remarks to the Author):

I'd like to thank the authors again for accommodating my requests. They have now made sufficient changes to the manuscript so that I believe further requests would be importunate. The study provides us important insights to the prefrontal circuitry in the brain and is a significant contribution to the neuroscience community.

NCOMMS-18-03143B

Riley et al.

Response to Reviewers

Reviewer #3:

I'd like to thank the authors again for accommodating my requests. They have now made sufficient changes to the manuscript so that I believe further requests would be importunate. The study provides us important insights to the prefrontal circuitry in the brain and is a significant contribution to the neuroscience community.

Response: Thank you.